

# Optical and microphysical properties of natural mineral dust and anthropogenic soil dust near dust source regions over Northwestern China

Xin Wang[1], Hui Wen[1], Jinsen Shi[1], Jianrong Bi[1], Zhongwei Huang[1], Beidou Zhang[1], Tian Zhou[1], Kaiqi Fu[1], Quanliang Chen[2], and Jinyuan Xin[3]

[1] Key Laboratory for Semi-Arid Climate Change of the Ministry of Education, College of Atmospheric Sciences, Lanzhou University, Lanzhou 730000, China

[2] Plateau Atmospheric and Environment Laboratory of Sichuan Province, College of Atmospheric Sciences, Chengdu University of Information Technology, Chengdu 610225, China

[3] State Key Laboratory of Atmospheric Boundary Layer Physics and Atmospheric Chemistry (LAPC), Institute of Atmospheric Physics, Chinese Academy of Sciences, Beijing 100029, China

*Correspondence to:* X. Wang (wxin@lzu.edu.cn)


**Abstract.**

From 3 April to 16 May 2014, a ground-based mobile laboratory was deployed to measure the optical and microphysical properties of tropospheric aerosols near dust source regions in Wuwei, Zhangye, and Dunhuang along the Hexi Corridor over Northwestern China. This study is novel in that we not only captured natural mineral dust near the Taklimakan and Badain Jaran desert regions but also characterized the properties of anthropogenic soil dust produced by agricultural cultivations (e.g., land planning, ploughing, and disking), especially during floating dust episodes. In this dust campaign, the aerosol scattering (absorption) coefficient ($\sigma_{sp}$ and $\sigma_{ap}$), single scattering albedo ($\omega$), scattering Ångström exponent ($Å_{sp}$), mass scattering efficiency (MSE), and aerosol size distribution were observed at 5-min intervals. The results indicate that large differences were found between the optical and microphysical properties of anthropogenic and natural dust because of floating dust episodes and dust storms. The values of $\sigma_{sp}$, $\sigma_{ap}$ and $\omega$ of PM$_{2.5}$ measured at 550 nm range from ~37–532 Mm$^{-1}$, ~2.2–55.3 Mm$^{-1}$ and ~0.64–0.98, respectively, because of the presence of anthropogenic soil dust during floating dust episodes, and the corresponding values under background conditions are ~21–163 Mm$^{-1}$, ~1.3–34.8 Mm$^{-1}$, and ~0.70–0.98, respectively. We note that the higher values of $\sigma_{sp}$ and $\omega$ and lower values of $Å_{sp}$ indicate that the aerosol particles observed during a strong dust storm in Zhangye were dominated by coarse mode particles originating from desert regions. However, the highest values of MSE reveal that the anthropogenic soil dust produced by agricultural cultivations can scatter more solar radiation than coarse mode particles.



# 1    Introduction

Mineral dust is one of the major types of tropospheric aerosols in the atmosphere
(Arimoto et al., 2006; Ramanathan et al., 2001). Mineral dust has a profound impact on
the radiative balance of the Earth by scattering and absorbing solar radiation (Huang et
al., 2010; Wang et al., 2010; Huang et al., 2014; Li et al., 2016); this dust can also act
as cloud condensation nuclei (CCN) to alter the precipitation rate and hydrological
cycle of the Earth (Rosenfeld et al., 2001). East Asia includes the Taklimakan, Tengger,
Badain Jaran and Gobi Deserts and is thus considered to be one of the major source
regions of natural dust in the world, as it produces large amounts of natural mineral dust
(Zhang et al., 1997; Wang et al., 2008; Che et al., 2011, 2013; Ge et al., 2014; Xin,
2005, 2010, 2015). Once dust aerosols are lifted into the atmosphere by strong surface
winds, they can be transported up to thousands of kilometres away from their source
regions (Chen et al., 2013; Liu et al., 2015; Huang et al., 2008). The long-range
transport of tropospheric aerosols from their dust source regions in East Asia plays a
key role in aerosol radiative forcing (Ge et al., 2011). For instance, Washington et al.
(2003) demonstrated that there is large-scale uncertainty associated with the estimation
of radiative forcing caused by dust particles, which is mainly due to their non-sphericity
and chemical composition. Compared to natural mineral dust, black carbon (BC), which
is generated from the incomplete combustion of fossil fuels and biomass burning, is
also a major anthropogenic pollutant. Numerous modelling studies have
demonstrated the importance of BC in regional climate change (Ramanathan et
al., 2005; Menon et al., 2002). Therefore, the largest source of uncertainty in





determining global radiative forcing is the quantification of the direct and indirect

effects of mineral dust and other anthropogenic aerosols (IPCC, 2013).

Recently, the potential impacts of anthropogenic soil dust have also received an

increasing amount of attention (Prospero et al., 2002; Huang et al., 2015a; Tegen and

Fung, 1995; Tegen et al., 2002; Shi et al., 2015; Pu et al., 2015; Wang et al., 2015).

Anthropogenic mineral dust can also influence air quality and human health through

the processes of their emission, transport, removal, and deposition (Aleksandropoulou

et al., 2011; Chen et al., 2013; Huang et al., 2014, 2015a, 2015b; Kim et al., 2009; Li

et al., 2009; Mahowald and Luo, 2003; Zhang et al., 2005, 2015). Ginoux et al. (2010)

estimated that anthropogenic dust accounts for 25% of all dust aerosols using

observational data from MODIS (Moderate Resolution Imaging Spectroradiometer)

Deep Blue satellite products combined with a land-use fraction dataset. Anthropogenic

dust primarily originates from urban and regional sources, especially during the winter;

this dust is commonly enriched in heavy metals and other toxic elements (Kamani et

al., 2015; Li et al., 2013; Wang et al., 2015; Zhang et al., 2013). Northeastern China

and its surrounding regions are generally regarded as industrial areas that are most

strongly affected by human activities. Because anthropogenic dust emissions from

disturbed soils are not well constrained, we define anthropogenic dust as mineral dust

from areas that have been disrupted by human activities, such as deforestation,

overgrazing, and agricultural and industrial activities (Aleksandropoulou et al., 2011;

Tao et al., 2014, 2015, 2017; Tegen and Fung, 1995; Tegen et al., 2002, 2004;

Thompson et al., 1988); anthropogenic dust is thus different than natural mineral dust,



which originates from desert regions (Che et al., 2011, 2013; Goudie and Middleton, 2001; Li et al., 2012; Park and Park, 2014; Pu et al., 2015; Wang et al., 2008, 2010). This assumption is consistent with the results of a recent study by Huang et al. (2015a), who found that anthropogenic dust comprises 91.8% of regional emissions in eastern

China and 76.1% of regional emissions in India (e.g., Figure 10 in Huang et al., 2015a). This may be due to the larger population densities of eastern China and India, which are characterized by intense human activity (Huang et al., 2015a; Wang et al., 2013). Understanding the microphysical and optical properties of natural dust mixed with the other atmospheric aerosols produced by human activities in the troposphere has a

critical impact on our ability to predict atmospheric compositions and global climate change (Nie et al., 2014; Ramanathan et al., 2007; Spracklen and Rap, 2013). Several attempts have been made to investigate the significance of the effects of dust on global climate, meteorology, atmospheric dynamics, ecosystems and human health (Rosenfeld et al., 2011; Qian et al., 2004). Limited field campaigns have focused on the properties

of natural dust and anthropogenic aerosols, especially those of the anthropogenic dust aerosols produced by human activities near dust source regions. In this paper, we focus on the measured optical and microphysical properties of natural dust and anthropogenic aerosols. This paper presents detailed emission information obtained from measurements made in Wuwei, Zhangye, and Dunhuang over Northwestern China from

3 April to 16 May in 2014 in order to better understand the sources of regional emissions and the mixing state of air pollution with mineral dust. We also used statistical analysis



to identify the possible signatures of natural dust storms transported from dust source regions.

## 2 Methodology

### 2.1 Measurement Site

The dust field campaign was carried out along the Hexi Corridor from 3 April to 16 May 2014. A ground-based mobile facility of the Semi-Arid Climate and Environment Observatory of Lanzhou University (SACOL) for Energy Atmospheric Radiation Measurements was used in three sites, which are located in Wuwei (37.72°N, 102.89°E; 3 - 7 April), Zhangye (39.04°N, 100.12°E; 9 - 28 April), and Dunhuang (39.96°N,

94.33°E; 3 - 16 May). The locations of these sites are shown in the top panel of Figure 1. The Hexi Corridor is considered to be a heavily polluted area because of the combination of local topography and the human activities occurring over Northwestern China. As shown in the bottom panels of Figure 1, the site in Wuwei is located only ~17 km away from the Tengger Desert and ~20 km away from the Qilian Mountains;

therefore, anthropogenic air pollutants originating from Wuwei City can directly influence the sampling site because of the prevailing wind direction along the local topography. The land surface type in Linze farmland in Zhangye (LFZ) is similar with Wuwei (Figure 2b). However, the sampling site in Dunhuang was located approximately 30 km away from the urban area and in the upwind direction of

Dunhuang, and the primary components in Dunhuang were dominated by natural mineral dust (Figure 2c). As shown in Figure 2a and 2b, the local tropospheric aerosols



in Wuwei and Zhangye were dominated by anthropogenic soil dust due to agricultural cultivation activity (e.g., land planning, ploughing, and disking), especially during the floating dust period.

## 2.2 Instrumentation

Automatic measurements of ambient temperature, relative humidity, pressure, wind direction, and wind speed were collected at 1-min intervals based on the microphysical and optical parameters of aerosols described above. Finally, all datasets measured during this field campaign were adjusted to standard temperature and pressure conditions (STP; T=273.15 K, P=101.325 kPa). We also measured aerosol absorption

and scattering coefficients, mass concentrations, and aerosol number size distribution during the 2014 dust field campaign, in which 5-min averaged data were used. The wind direction datasets are associated with the aerosol absorption and scattering coefficients and can be used to determine the origins of natural and anthropogenic dust. Figure 3a shows the ground-based mobile laboratory used in Dunhuang during the dust

field campaign along the Hexi Corridor over Northwest China. A sampler inlet was installed at the top of the ground-based mobile laboratory, using 1 $\mu$m and 2.5 $\mu$m impactors for different instruments, which are shown in Figure 3b. All of the collectors were operated at 50°C to dry the aerosols (i.e., to a relative humidity (RH) of less than 40%). The mass concentration of $PM_{2.5}$ was measured continuously using an R&P

1400a analyser, which is based on the principle of tapered element oscillating microbalance (TEOM), with a flow rate of 16.7 L min$^{-1}$. To compare the properties of



coarse and fine mode particles, two integrating nephelometers (TSI Model 3563) with

1 $\mu$m and 2.5 $\mu$m impactors were employed to measure aerosol scattering coefficients

($\sigma_{sp}$) at 450, 550, and 700 nm; the detection limits of these instruments were $0.44 \times 10$

Mm$^{-1}$, $0.17 \times 10$ Mm$^{-1}$, and $0.26 \times 10$ Mm$^{-1}$, respectively, and the instruments had a

signal-to-noise ratio (S:N) of 2:1 (Anderson et al., 1996; Shi et al., 2013). Then, a multi-

angle absorption photometer (MAAP-5012) was used to measure aerosol absorption

coefficients ($\sigma_{ap}$) at 670 nm. The particle size distribution ranging from 0.5–20 $\mu$m was

measured using an aero-dynamical particle sizer (APS-3321), assuming that all aerosols

are homogeneous and spherical particles, despite the fact that the observed coarse mode

dust particles exhibit non-spherical geometries (Mishchenko et al., 1995). The mass

absorption coefficient (MAC, hereinafter $\alpha$) is a key parameter that can be used to

attribute the light absorption of aerosols to BC and to understand its effects on climate.

The MAC values of BC span a wide range in the literature. For instance, the MAC of

BC has been defined as 7.5 m$^2$ g$^{-1}$ at 550 nm and 12–13 m$^2$ g$^{-1}$ at ~350 nm (Adler et al.,

2010); these values are calculated by assuming that the imaginary part of the complex

refractive index (RI) of BC is independent of the wavelength (λ) (Bond and Bergstrom,

2006). To calculate the aerosol absorption coefficient ($\sigma_{ap}$), the following equation is

used:

$$\sigma_{ap} = \alpha \times m_{BC} \tag{1}$$

where $\alpha = 6.6$ m$^2$ g$^{-1}$ and $m_{BC}$ is the mass concentration of BC.

The wavelength-dependent variation of $\sigma_{sp}$ is characterized by the scattering

Ångström exponent (SAE, $Å_{sp}$), which is defined as:





$$\mathring{A}_{sp}(\lambda_1/\lambda_2) = -\frac{\ln(\sigma_{sp,\lambda_1}/\sigma_{sp,\lambda_2})}{\ln(\lambda_1/\lambda_2)} \qquad (2)$$

where $\sigma_{sp,\lambda_1}$ and $\sigma_{sp,\lambda_1}$ are the aerosol scattering coefficients at wavelengths $\lambda_1$ and $\lambda_2$, respectively. In this paper, we calculated $\mathring{A}_{sp}$ from 450–700 nm (i.e., from $\sigma_{sp,450}$ and $\sigma_{sp,700}$).

The attribution of observed atmospheric light absorption to BC is an important step in understanding the overall climate effects of aerosols. Some studies have attempted to perform this attribution based on the assumption of the wavelength dependence of absorption (e.g. Favez et al., 2009; Yang et al., 2009). It is often assumed that the imaginary RI of BC is independent of wavelength ($\lambda$) and that the absorption cross-

section of BC varies as $\lambda^{-1}$ (Bond and Bergstrom, 2006).

To calculate the single scattering albedo ($\omega$) at 550 nm, we first adjust the $\sigma_{ap}$ values from 670 to 550 nm; this equation is listed as follows:

$$\sigma_{ap,550} = \sigma_{ap,670} \times \left(\frac{\lambda_{670}}{\lambda_{550}}\right)^{AAE} \qquad (3)$$

where $\lambda$ is wavelength and the absorption Ångström exponent (AAE) corresponds to

the $\lambda^{-1}$ dependence of the absorption of BC, which is typically assumed to be AAE=1. However, we note that the actual AAE of BC can be greater or less than 1, as this value is highly dependent on the internal/external core size and its ageing process (Gyawali et al., 2009).

The single scattering albedo is a key parameter that can be used to investigate the

microphysical and optical properties of atmospheric aerosols. The single scattering albedo is defined as the ratio of the scattering coefficient to the total extinction coefficient (i.e., the scattering + absorption coefficients) at 550 nm using Eq. (4):





$$\omega_{550} = \frac{\sigma_{sp}}{\sigma_{ap}+\sigma_{sp}} \qquad (4)$$

The parameter of mass scattering efficiency (MSE) is calculated as the slope of the

RMA linear regression of $\sigma_{sp}^{2.5}$ and PM$_{2.5}$, which is calculated using Eq. (5):

$$MSE = \frac{\sigma_{sp}^{2.5}}{PM_{2.5}} \qquad (5)$$

5    where $\sigma_{sp}^{2.5}$ is the aerosol scattering coefficient at 550 nm and PM$_{2.5}$ is the mass

concentration of atmospheric aerosols with size diameters of less than 2.5 $\mu$m.

## 3    Results

During this dust field campaign, four floating dust episodes (which are shown as dotted

boxes in Figure 4) occurred on 3–7 April in Wuwei and on 9–12, 14–15 and 25–28

10    April in Zhangye. We also observed the microphysical and optical properties of natural

mineral dust during a heavy dust storm (shown as a solid box in Figure 4) from 23 to

25 April 2014. According to the land surface types shown in Figure 2, one of the major

aims of this study is to investigate the characteristics of anthropogenic and natural dust,

which are present in floating dust episodes and dust storms, respectively, and represent

15    the fine mode soil dust that is produced by agricultural cultivations and the coarse mode

mineral dust that originates from desert regions, respectively. Therefore, significant

differences in the optical and microphysical properties of these aerosols were observed

under different atmospheric conditions (e.g., dust storms, floating dust episodes and

background conditions). Figure 4 illustrates the temporal variations in $\sigma_{ap}$, $\sigma_{sp}$, $\omega$

20    and MSE at 550 nm as well as those in Å$_{sp}$ (calculated from 450 nm to 700 nm) and

aerosol size distribution in Wuwei, Zhangye, and Dunhuang from 3 April to 16 May





2014. These surveys were conducted in Wuwei, Zhangye, and Dunhuang in chronological order, as is illustrated in Figure 4. The statistical analyses of the optical parameters measured during this dust field campaign are summarized in Table 1 (hereinafter, these results are given as the mean ± the standard deviation of the 5-min

averaged datasets). One of the most significant features in Figure 4a is that the variation of $\sigma_{sp}^{2.5}$ is highly consistent with that of $\sigma_{sp}^{1.0}$ during the field campaign. The values of $\sigma_{sp}^{1.0}$ and $\sigma_{sp}^{2.5}$ are very close in Wuwei and Zhangye; however, the large differences observed in Dunhuang indicate that fine mode particles dominate the scattering coefficient in farmland regions, whereas coarse mode particles play a more important

role closer to the desert regions of Dunhuang. Except for the values obtained during a heavy dust storm in Zhangye, the $\sigma_{sp}^{2.5}$ values measured at 550 nm using the nephelometer range from ~50–429 Mm$^{-1}$ and ~20–532 Mm$^{-1}$ at Wuwei and Zhangye, respectively, which can be compared to the $\sigma_{ap}^{2.5}$ values of ~3.6–69.6 Mm$^{-1}$ and ~1.3– 64.5 Mm$^{-1}$ measured at Wuwei and Zhangye, respectively. The lowest value of $\sigma_{ap}^{2.5}$

obtained during the field campaign (i.e., 2.7 ± 1.2 Mm$^{-1}$) was collected in Dunhuang. This observation reveals that natural mineral dust is still a weaker absorber than anthropogenic soil dust that has been mixed with air pollutants (e.g., BC). Values of $\omega_{550}$ were estimated using the aerosol scattering coefficient obtained from the nephelometer and the aerosol absorption coefficient of the MAAP at 550 nm, with a

nominal instrumental uncertainty of ± 0.02. Compared with Figure 4a, Figure 4b indicates that the majority of $\omega_{550}$ values are much higher in Dunhuang than they are in the other two sites; these values range from ~0.80 to 0.99, with a mean value of





0.95 ± 0.02. Similar results were also found in Zhangye during the dust storm because

of the presence of coarse mode particles. However, only 0.7% and 21.9% of the values

of $\omega_{550}$ reach up to 0.95 in Wuwei and Zhangye (Figure 7), and their average $\omega_{550}$

values are much lower (0.91 ± 0.03 and 0.93 ± 0.04, respectively) and exhibit larger

variation than those in Dunhuang. This phenomenon most likely indicates that natural

dust aerosols are dominant in Dunhuang, since mineral dust scatters more and absorbs

less than other atmospheric aerosols. Figure 4c reveals that the $Å_{sp}$ values of aerosols

with diameters of less than 1 $\mu$m ($Å_{sp}^{1.0}$) are much higher than those of $Å_{sp}^{2.5}$ and that

$Å_{sp}$ values decrease significantly from Wuwei to Dunhuang. The average MSE values

in Wuwei, Zhangye and Dunhuang are 2.8 ± 0.7 $m^2 g^{-1}$, 2.2 ± 0.8 $m^2 g^{-1}$ and 1.5 ± 0.7

$m^2 g^{-1}$, respectively, which have 5-min maximum values of 6.3 $m^2 g^{-1}$, 9.5 $m^2 g^{-1}$, and

8.5 $m^2 g^{-1}$, respectively. The higher MSE values in Wuwei and Zhangye reflect the fact

that anthropogenic dust, which is influenced by local soil dust during floating dust

episodes, scatters more solar radiation than natural dust (Figure 4d). Figure 4e indicates

that fine mode particles that are less than 1 $\mu$m in diameter are dominant in Wuwei and

Zhangye, except during a strong dust storm that occurred in Zhangye on 23–25 April

2014. Although the number concentration of fine mode particles is also higher than that

of coarse mode particles in Dunhuang, the average percentage of coarse mode particles

relative to total atmospheric particles is 55%, which is higher than the relative

percentages observed in Wuwei and Zhangye. On 23–25 April 2014, a severe dust

storm occurred in Zhangye, along with a strong northerly wind. The 5-min average

$\sigma_{sp}^{2.5}$ value at 550 nm increased to 5813 $Mm^{-1}$, which is ~10 times higher than that





measured in non-dust plume periods in Zhangye (532 Mm$^{-1}$), while the maximum value

of $\sigma_{ap}^{2.5}$ was 59.6 Mm$^{-1}$ during this dust storm, which is slightly lower than that

measured during non-dust plume periods (64.5 Mm$^{-1}$). Figures 4b and 4c indicate that

the peaks of $\omega_{550}$ (>0.99) that are associated with the lowest values of Å$_{sp}$ are close

to those observed in another field campaign that studied dust dominant particles in

Northwestern China (Li et al., 2010). Figure 4e demonstrates that the number

concentration of coarse mode particles with diameters of 1–5 $\mu$m is higher than 900

cm$^{-3}$, which indicates that pure coarse mode particles from desert regions are dominant

during dust storms in Zhangye. These results are consistent with those of a previous

study, in which the aerosol diameter (De) of PM$_{10}$ was determined to be larger during

dust plume periods than it was during non-dust plume periods (Wang et al., 2010).

Here, we also present the diurnal cycles of $\sigma_{sp}^{2.5}$, $\sigma_{sp}^{1.0}$, $\sigma_{ap}^{2.5}$, and $\omega_{550}$, as well as those

of the Å$_{sp}$ and MSE values in Wuwei (red line), Zhangye (black line), and Dunhuang

(blue line) throughout these periods (Figure 5). As shown in Figure 5a and 5b, the

values of $\sigma_{sp}^{2.5}$, $\sigma_{sp}^{1.0}$ and $\sigma_{ap}^{2.5}$ at 550 nm present bimodal distributions in Wuwei and

Zhangye, which are consistent with the variations in $\omega_{550}$ (Figure 5e). The average

values of $\sigma_{sp}^{2.5}$, $\sigma_{sp}^{1.0}$ and $\sigma_{ap}^{2.5}$ in Dunhuang are 53 Mm$^{-1}$, 25 Mm$^{-1}$ and 2.6 Mm$^{-1}$,

respectively, which are much lower than those measured in Wuwei and Zhangye

(Figure 5a and 5b). The highest values of $\sigma_{ap}^{2.5}$ indicate that not only anthropogenic

mineral dust but also local air pollutants (e.g., BC and OC) were found in these regions;

these pollutants originated from biomass and the burning of fossil fuels. Compared with

the lower $\omega_{550}$ values that occurred at ~8:00 and ~20:30 LST in Wuwei and Zhangye,



the presence of only slight variations in $\omega_{550}$ in Dunhuang indicates that coarse mode

particles (e.g., natural dust aerosols) are dominant in Dunhuang (Figure 5e).

Additionally, a comparison of the Å$_{sp}$ values due to fine mode and coarse mode particles

that are less than 1 $\mu$m and 2.5 $\mu$m in size, respectively, indicates that there are large

variations in Å$_{sp}^{2.5}$ and Å$_{sp}^{1.0}$ in Wuwei, ranging from ~1.0–1.7 and ~1.8–2.3,

respectively. The lowest values of Å$_{sp}^{2.5}$ and Å$_{sp}^{1.0}$ observed in Dunhuang also suggest

that its atmospheric aerosols are dominated by coarse mode particles (Figure 5c-5d).

Another feature is that the Å$_{sp}^{1.0}$ value of fine mode particles is significantly higher than

that of Å$_{sp}^{2.5}$. Large variations in MSE are found at all three sites; MSE values are

highest during the floating dust episodes in Wuwei and lowest in Dunhuang because of

the amounts of pure coarse mode particles that are present during background surface

conditions (Figure 5f). Diurnal variations in the aerosol number distribution (hereinafter

defined as dN/dlogD$_p$) in the range of 0.5 $\mu$m $\leq$ D$_p$ (particle diameter) $\leq$ 5 $\mu$m were

also observed using the aerodynamic particle sizer (APS) instrument (Figure 6). As is

shown in Figure 6a, the accumulated fine mode particles (<1 $\mu$m) increased in Wuwei,

yielding a maximum number distribution of more than 300 cm$^{-3}$ due to the frequent

outbreaks of floating dust episodes that occurred in Wuwei on 3–7 April 2014. The

similar patterns of fine mode particles imply that Wuwei has very similar pollutant

emission sources as Zhangye, but with a slightly lower number distribution of fine mode

particles, as is shown in Figure 6b. We suggest that the fine mode particles represent

the dominant contributions in Wuwei and Zhangye, which is due to the formation of

local anthropogenic soil dust by agricultural cultivations. However, large differences





are found in Dunhuang because of its higher percentage of coarse mode particles relative to total atmospheric particles; this region yields 5-min average values of 203 ± 125 cm$^{-3}$ and 234 ± 248 cm$^{-3}$ for fine mode and coarse mode particles, respectively (Figure 6c).

Figure 7 shows the histograms of the single scattering albedo values observed at 550 nm from 3 April to 16 May 2014. During the floating dust period in Wuwei, the majority of the $\omega_{550}$ values of fine mode particles that originated from anthropogenic soil dust range from ~0.90–0.93; approximately 10%–20% of those values range from ~0.88–0.90 and ~0.93–0.95. The overall range of $\omega_{550}$ values observed in Zhangye is similar

to that observed in Wuwei. The $\omega_{550}$ values that range from ~0.90–0.93 are 30% higher than those in Zhangye. This observation indicates that the atmospheric aerosols not only include anthropogenic soil dust that is smaller than 1 $\mu$m but also have undergone mixing with air pollutants (e.g., BC) during their transportation from urban and industrial regions. This result is consistent with that of Li et al. (2010), who noted

that the SSA of a dust storm was approximately 0.98 for coarse mode particles, while lower SSA values (i.e., ranging from 0.89 to 0.91) were closely related to local air pollution. Thus, we infer that the occurrence of much lower $\omega_{550}$ values in Wuwei and Zhangye than in Dunhuang is due to the mixing of anthropogenic pollutants with local mineral dust. However, the $\omega_{550}$ values in Dunhuang range from ~0.93–0.98,

with the majority of these values falling between ~0.95–0.98 because of the high percentage of coarse mode particles. These results are consistent with that of a previous study, which indicated that the surface measurement of SSA for coarse mode particles



from Saharan desert regions at 550 nm yielded a value of $0.97 \pm 0.02$ (Cattrall et al., 2003). Additionally, the wind roses in Figure 8 can be used to provide further insights into the correlation between meteorology and the aerosol scattering (absorption) coefficients. The wind direction accompanying $\sigma_{sp}^{2.5}$ and $\sigma_{ap}^{2.5}$ at 550 nm most likely

represents the emissions from both local sources and regional transport from remote regions. The dominant wind direction is generally more abundant to the west. Figures 8a and 8b indicate that the higher values of $\sigma_{sp}^{2.5}$ are found along with the northwest wind, while the majority of $\sigma_{ap}^{2.5}$ values are dominated by the southeast wind because of the emissions of anthropogenic pollutants from the city centre of Wuwei. However,

the fact that the highest values of $\sigma_{sp}^{2.5}$ and $\sigma_{ap}^{2.5}$ are associated with the western wind in Zhangye most likely indicates that these emissions originate from both anthropogenic soil dust and air pollutants from their upwelling regions (Figure 8c and 8d). However, both $\sigma_{sp}^{2.5}$ and $\sigma_{ap}^{2.5}$ were probably influenced by coarse mode mineral dust due to the presence of northwest wind in Dunhuang (Figure 8e and 8f).

MSE is a key parameter that can be used to estimate the radiative forcing effects due to atmospheric particles on global climate. Therefore, several studies have been performed to determine the optical properties of aerosols using MSE values (Laing et al., 2016). For instance, Hand and Malm (2007) noted that the MSE is mainly dependent on particle composition (e.g., the particle refractive index and aerosol size distribution).

As shown in Figure 9a, coarse mode particles have significantly higher $\omega_{550}$ (>0.93) and lower MSE (1< MSE<2) values because of the presence of natural mineral dust under background weather conditions in Dunhuang. However, there appears to be no





clear difference between the $\omega_{550}$ and MSE values due to floating dust periods in Wuwei and Zhangye. The presence of lower and higher MSE values in Wuwei and Zhangye, respectively, suggests that fine mode particles can not only be attributed to floating dust periods (due to local soil dust) but also include BC, OC and other air

pollutants that originated from the burning of biomass and fossil fuels. For instance, the large variations in $\omega_{550}$ (~0.71–0.95 and ~0.68–0.99, respectively) and MSE (~0.9–6.3 m$^2$ g$^{-1}$ and ~0.4–9.5 m$^2$ g$^{-1}$, respectively) observed in Wuwei and Zhangye are consistent with values that were previously measured during dust storms or biomass burning events (Li et al., 2010; Laing et al., 2016). Another notable feature is the

remarkable discrepancy between the optical properties of aerosols for a given type of aerosol with diameters of less than 1 $\mu$m and 2.5 $\mu$m. Although the values of $\sigma_{sp}^{1.0}$ measured during this dust field campaign are only slightly lower than those of $\sigma_{sp}^{2.5}$ (as is indicated in Figures 9b and 9c), the $\text{Å}_{sp}^{1.0}$ values range from ~1.1–2.4 (mean: 2.1) for fine mode particles because of floating dust episodes in Wuwei, compared to the values

of ~0.2–1.7 (mean: 1.3) observed during the same period in Wuwei. Similar results are also found at the other two sites in Zhangye and Dunhuang.

The microphysical and optical properties of aerosols observed during this dust field campaign are summarized in Figure 10. The boxes and whiskers denote the 10$^{th}$, 25$^{th}$, median, 75$^{th}$, and 90$^{th}$ percentiles of the data, with dots marking their average values.

We determined that the values of both $\sigma_{sp}$ and $\sigma_{ap}$ at 550 nm are higher in Wuwei during floating dust episodes than that in Zhangye and Dunhuang during background conditions. The average values of $\sigma_{sp}^{1.0}$, $\sigma_{sp}^{2.5}$, and $\sigma_{ap}^{2.5}$ measured at 550 nm during





this field campaign in Wuwei are $75 \pm 29$ Mm$^{-1}$, $102 \pm 40$ Mm$^{-1}$, and $11.5 \pm 7.8$ Mm$^{-1}$, respectively. The 90[th] percentage of $\sigma_{sp}^{2.5}$ in Zhangye is due to the dust storm that occurred on 23–25 April 2014. However, the highest $\omega_{550}$ value is found in Dunhuang and is due to the presence of coarse mode particles during background conditions. We

note that there are large differences in Å$_{sp}$ between fine mode and coarse mode particles. The average values of Å$_{sp}^{2.5}$ and Å$_{sp}^{1.0}$ are $1.3 \pm 0.3$ and $2.1 \pm 0.2$ during floating dust episodes in Wuwei, respectively, whereas those in Dunhuang are $0.5 \pm 0.3$ and $0.9 \pm 0.4$, respectively. The highest MSE in Wuwei and the lowest MSE in Dunhuang indicate that fine mode mineral dust particles scatter more solar radiation than coarse

mode particles. Aerosol optical depth (AOD) is a major optical parameter for aerosol particles and is a key factor affecting global climate (Holben et al., 1991, 2001, 2006; Srivastava and Bhardwaj, 2014). AOD can not only represent local air pollution but can also be used to observe dust storms. For instance, Dubovik et al. (2002) demonstrated that non-spherical mineral dust can be retrieved using the assumption of spherical

aerosols for high aerosol loading (AOD >0.5, Å <0.7) in desert regions due to dust events. Figure 11 illustrates the spatial distribution of deep blue AOD at 550 nm in East Asia retrieved using Terra MODIS during a heavy dust storm over northern China on 24 April 2014. During this dust storm, the spatial distribution of high aerosol loadings with AOD values of >1.6 over Northwest China was observed; in this distribution, the

transport of natural mineral dust from the Taklimakan Desert to the downwelling regions over China can be clearly seen. The most prominent feature in Figure 12 is that $\sigma_{sp}^{2.5}$ reaches its peak value from ~209 Mm$^{-1}$ to 5813 Mm$^{-1}$ and that a strong relationship





($R^2$=0.91) existed between $\sigma_{sp}^{2.5}$ and $\sigma_{ap}^{2.5}$ during the dust storm on 23–25 April 2014. However, the values of $\sigma_{ap}^{2.5}$ observed during the dust storm are consistent with those measured during floating dust episodes in Wuwei and Zhangye. Therefore, we note that large differences in $\sigma_{sp}^{2.5}$ are found between natural and anthropogenic mineral dust

because of the presence of fine mode and coarse mode particles during floating dust episodes and dust storms, but that they record similar ranges of $\sigma_{ap}^{2.5}$. We also observed the lowest values of $\sigma_{ap}^{2.5}$ and $\sigma_{sp}^{2.5}$ (which range from ~0.9–13.1 and ~14–264 Mm$^{-1}$, respectively) in Dunhuang on 3–16 May, which indicates that natural mineral dust represents the dominant particles under background conditions. The observed higher

$\sigma_{ap}^{2.5}$ and lower $\sigma_{sp}^{2.5}$ values suggest that fine mode anthropogenic dust particles were dominant in Wuwei and Zhuangye during floating dust episodes, in contrast to the coarse mode particles observed during the dust storm in Dunhuang.

Figure 13 shows the average aerosol number size distribution observed using the APS instrument during this field campaign. These data clearly show that the dominant

particles during the dust storm in Zhangye are coarse mode particles ranging in size from 1 $\mu$m to 5 $\mu$m, which reach a maximum value of 596 cm$^{-3}$ at a size of 1.60 $\mu$m. Compared with dust storms, the value of dN/dlogD$_p$ reaches a peak with values of ~335 cm$^{-3}$ and ~345 cm$^{-3}$ at 0.67 $\mu$m during typical floating dust episodes on 4–7 and 9–15 April 2014, respectively. This observation indicates that fine mode anthropogenic soil

dust mixed with local air pollutants was dominant during floating dust episodes on 3–7 and 9–15 April 2014 in Wuwei and Zhangye. However, another floating dust episode that occurred in Zhangye reveals a bimodal variation of dN/dlogD$_p$ at 0.67 $\mu$m and 1.49

$\mu$m. It also should be noted that the lowest value of dN/dlogD$_p$ for fine mode particles was observed in Dunhuang because of the background weather conditions. These results are very close to those of previous studies that stated that atmospheric particles were dominated by both anthropogenic soil dust and air pollutants during floating dust

episodes; however, the amount of coarse mode particles increased sharply during natural dust storms that originated from dust source regions (Wang et al., 2010; Li et al., 2010).

## 4    Conclusions

To determine the optical and microphysical properties of atmospheric particles in

anthropogenic soil dust and natural mineral dust, a ground-based mobile laboratory was deployed near the dust source regions along the Hexi Corridor over Northwest China from 3 April to 16 May 2014. Two of the sampling sites were located in farmland areas in Wuwei and Zhangye, and the other site was located near the edge of the Gobi Desert in Dunhuang. Therefore, the land surface types in Wuwei and Zhangye represent

anthropogenic soil dust generated by human activities (e.g., ploughing, coal combustion from domestic use, and biomass burning), while the land surface type in Dunhuang represents mineral dust.

During this dust field experiment, which was performed from April to May 2014, four floating dust episodes and one dust storm episode were observed in Wuwei and

Zhangye. There are two major findings observed in this study. The most prominent conclusion is that there are significant differences in the optical and microphysical




properties of aerosols between anthropogenic soil dust and natural mineral dust under different atmospheric conditions (e.g., dust storms, floating dust episodes and background conditions). For instance, the average values of $\sigma_{sp}^{2.5}$, $\sigma_{ap}^{2.5}$, $\omega_{550}$ at 550 nm, $\text{Å}_{sp}^{2.5}$, MSE and dN/dlogD$_p$ are 183 Mm$^{-1}$, 6.9 Mm$^{-1}$, 0.91, 0.9, 2.4 m$^2$ g$^{-1}$ and 2072

5   cm$^{-3}$, respectively, during floating dust periods, whereas these values are 1108 Mm$^{-1}$, 13.5 Mm$^{-1}$, 0.99, -0.02, 1.7 m$^2$ g$^{-1}$, and 9923 cm$^{-3}$, respectively, during dust storms. The number concentrations of coarse mode particles with diameters of 1–5 $\mu$m can reach a peak of 900 cm$^{-3}$, which reveals that pure coarse mode particles from desert regions were dominant during dust storms in Zhangye. However, the overall variations in $\omega_{550}$,

which ranges from 0.71 to 0.95 and 0.64 to 0.98 during floating dust episodes in Wuwei and Zhangye, respectively, indicate that atmospheric aerosols not only include anthropogenic soil dust that is smaller than 1 $\mu$m but has also undergone mixing with air pollutants (e.g., BC) because of their transportation from urban and industrial regions. In addition to the large discrepancies between fine mode and coarse mode

particles observed under different weather conditions, there are also significant differences between the optical and microphysical properties of the given atmospheric particles that are smaller than 1 $\mu$m and 2.5 $\mu$m. We note that the values of $\sigma_{sp}^{1.0}$ (75 $\pm$ 29 Mm$^{-1}$) are only slightly lower than those of $\sigma_{sp}^{2.5}$ (102 $\pm$ 40 Mm$^{-1}$) that are observed in Wuwei. However, there are significant differences between the values of

$\text{Å}_{sp}^{1.0}$ and $\text{Å}_{sp}^{2.5}$, which range from ~1.1–2.4 (mean: 2.1) and ~0.2–1.7 (mean: 1.3), respectively, because of the occurrence of floating dust episodes in Wuwei.

During the heavy dust storms studied here, atmospheric aerosols sharply increased

because of the addition of coarse mode particles ranging in diameter from 1–5 $\mu$m. The

maximum $dN/dlogD_p$ value for coarse mode particles reached 596 cm$^{-3}$ at 1.60 $\mu$m,

which is 5 times higher than that observed at the same site under background weather

conditions. However, $dN/dlogD_p$ reached its peak values of 335 cm$^{-3}$ and 345 cm$^{-3}$ for

fine mode particles at 0.67 $\mu$m during floating dust events in Wuwei and Zhangye,

respectively. These results indicate that these atmospheric aerosols not only were

dominated by anthropogenic soil dust produced by agricultural cultivations but also

underwent mixing with local air pollutants because of the burning of biomass and coal.

These results are very similar to those of previous studies, which indicated that

atmospheric particles were dominated by both anthropogenic soil dust and air pollutants

during floating dust events but that natural mineral dust due to dust storms originated

from dust source regions.

## 5 Data availability

All data sets and codes used to produce this study can be obtained by contacting Xin

Wang (wxin@lzu.edu.cn).The MODIS data used in this study are available at Aerosol

Product, https://modis.gsfc.nasa.gov/data/dataprod/mod04.php.

*Competing interests.* The authors declare that they have no conflict of interest.

***Acknowledgements.*** This research was supported by the Foundation for Innovative

Research Groups of the National Science Foundation of China (41521004), the

National Science Foundation of China under Grant 41522505, and the Fundamental





Research Funds for the Central Universities (lzujbky-2015-k01 and lzujbky-2016-k06).

The MODIS data were obtained from the NASA Earth Observing System Data and

Information System.





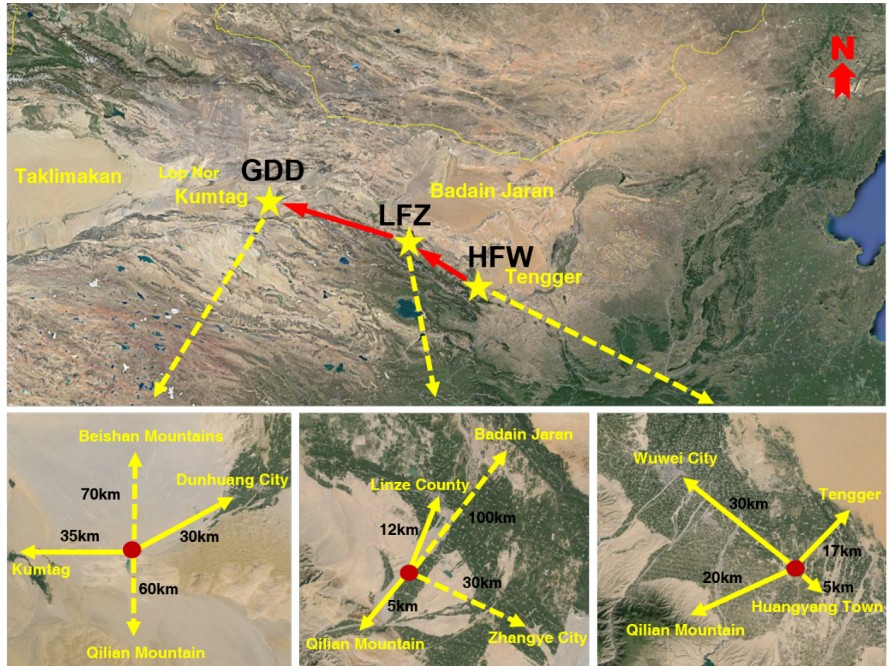

**Figure 1.** The sampling locations of ground-based mobile laboratory and their surrounding areas near dust source regions during the 2014 dust field campaigns at **(a)** Huangyang Farmland in Wuwei (HFW, 37.72°N, 102.89°E; 1691 m a.s.l.), **(b)** Linze Farmland in Zhangye (LFZ, 39.04°N, 100.12°E; 1578 m a.s.l.) and **(c)** Gobi Desert in Dunhuang (GDD, 39.96°N, 94.33°E; 1367 m a.s.l.).



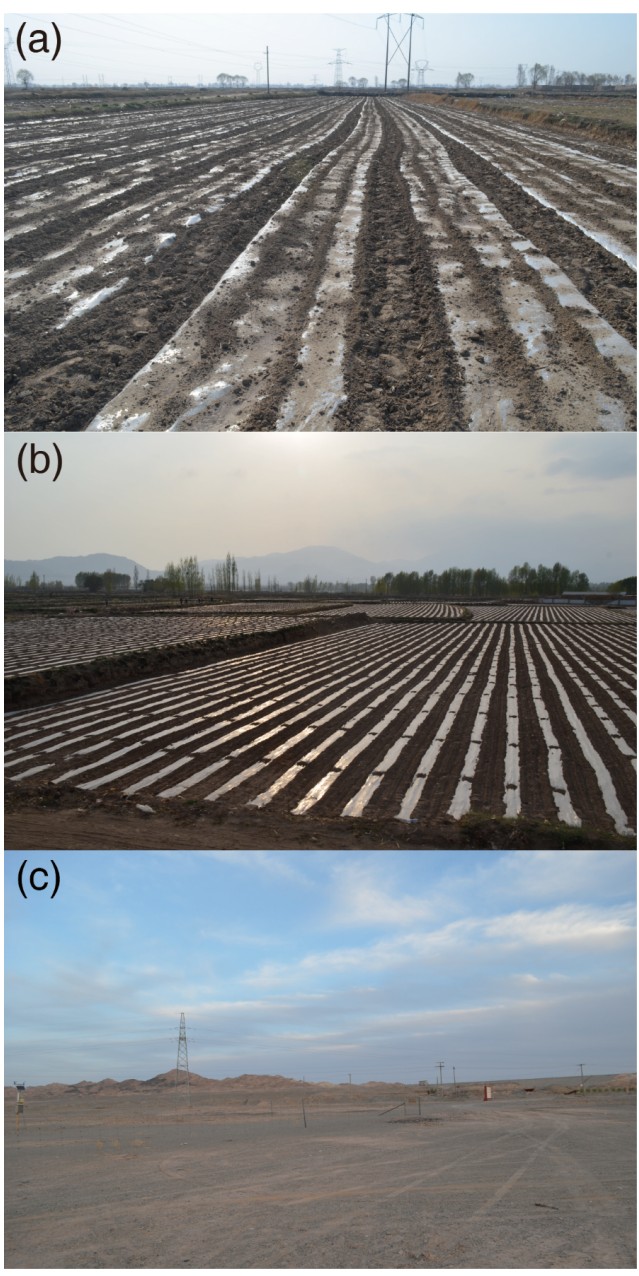

**Figure 2.** Same as **Figure 1** but for land surface conditions at **(a)** Huangyang Farmland in Wuwei (HFW), **(b)** Linze Farmland in Zhangye (LFZ), and **(c)** Gobi Desert in Dunhuang (GDD).





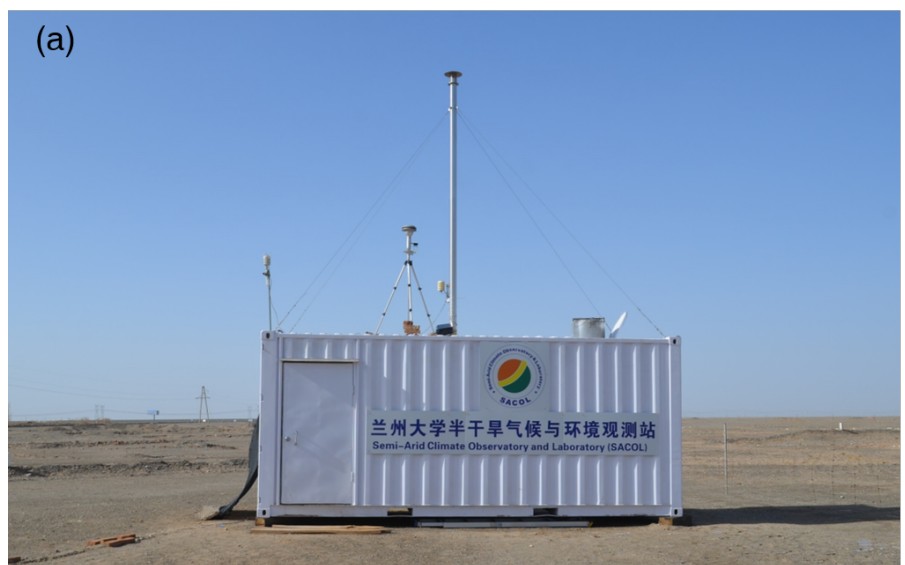

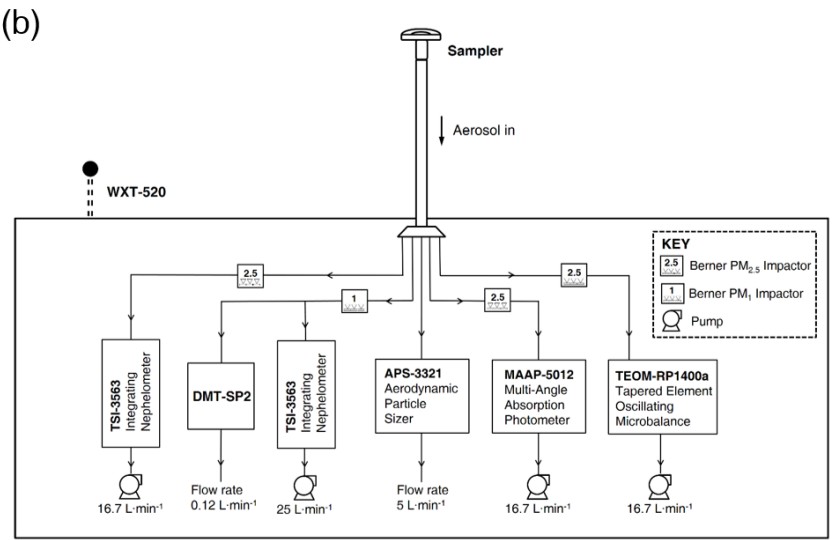

**Figure 3. (a)** The ground-based mobile laboratory in Dunhuang and **(b)** the schematic diagram of the ensemble instrumentation system.





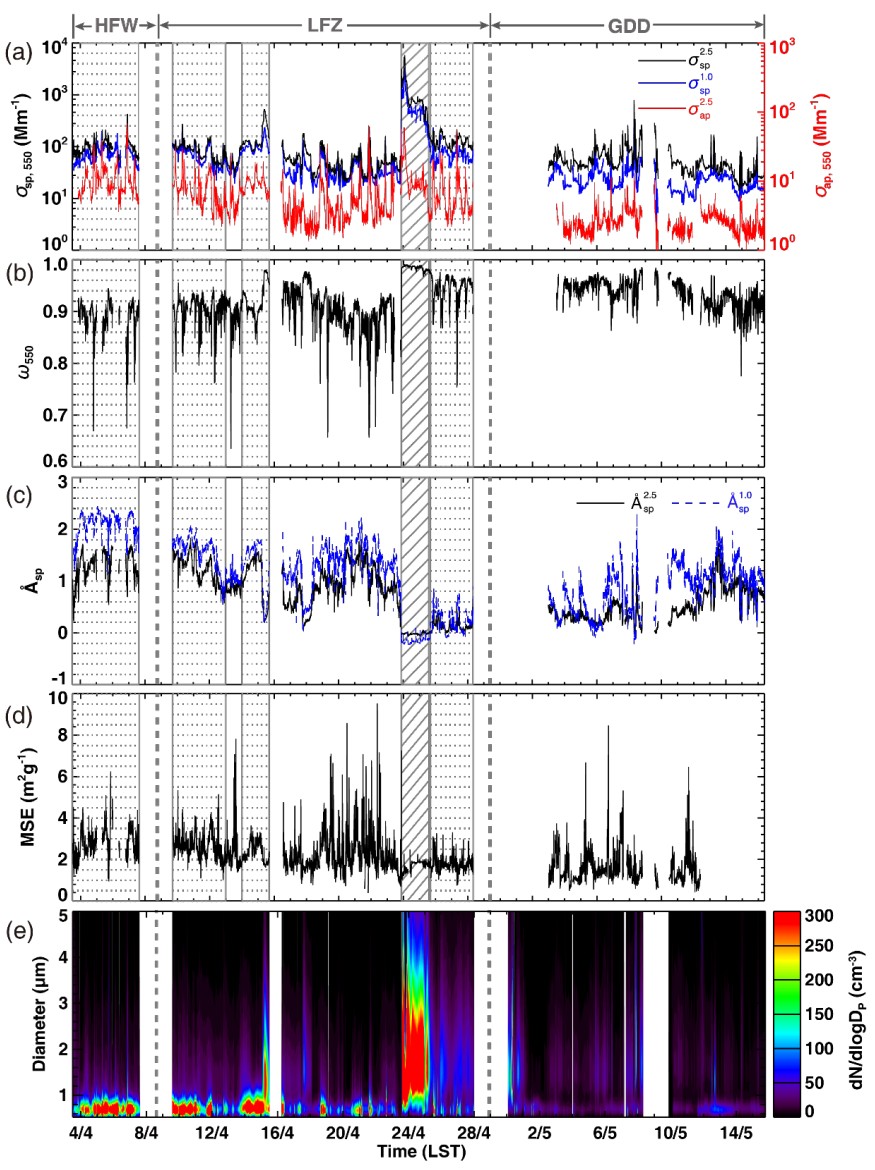

**Figure 4.** Temporal variations in **(a)** aerosol scattering (absorption) coefficient at 550 nm, **(b)** single scattering albedo at 550 nm, **(c)** scattering Ångström exponent (calculated from 450 nm to 700 nm), **(d)** mass scattering efficiency (MSE) at 550 nm, and **(e)** aerosol size distribution (dN/dlogD_p, 0.5 $\mu$m< D_p<5 $\mu$m) during the entire period from 3 April to 16 May 2014. The shaded box represents a strong dust storm that occurred in Zhangye, and the dotted boxes represent four floating dust episodes that occurred in Wuwei and Zhangye.





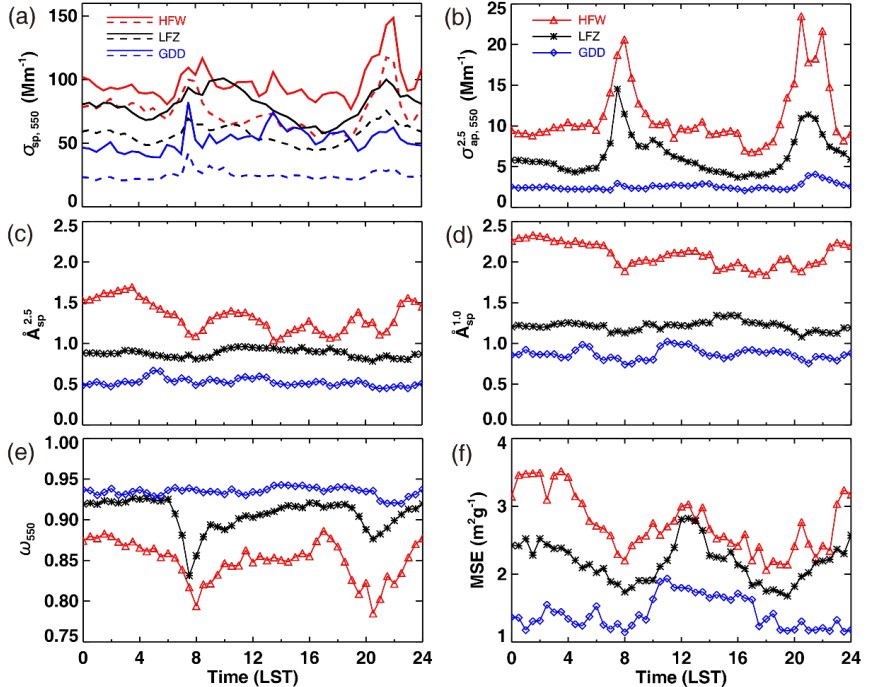

**Figure 5.** Diurnal variations in **(a)** aerosol scattering coefficient at 550 nm, where solid lines represent the variations in $PM_{2.5}$ and dotted lines represent the variations in $PM_{1.0}$; **(b)** the aerosol absorption coefficient and the scattering Ångström exponent for **(c)** $PM_{2.5}$ and **(d)** $PM_{1.0}$ (both calculated from 450 nm to 700 nm); **(e)** single scattering albedo at 550 nm; and **(f)** mass scattering efficiency (MSE) at 550 nm in Wuwei, Zhangye, and Dunhuang from 3 April to 16 May 2014.





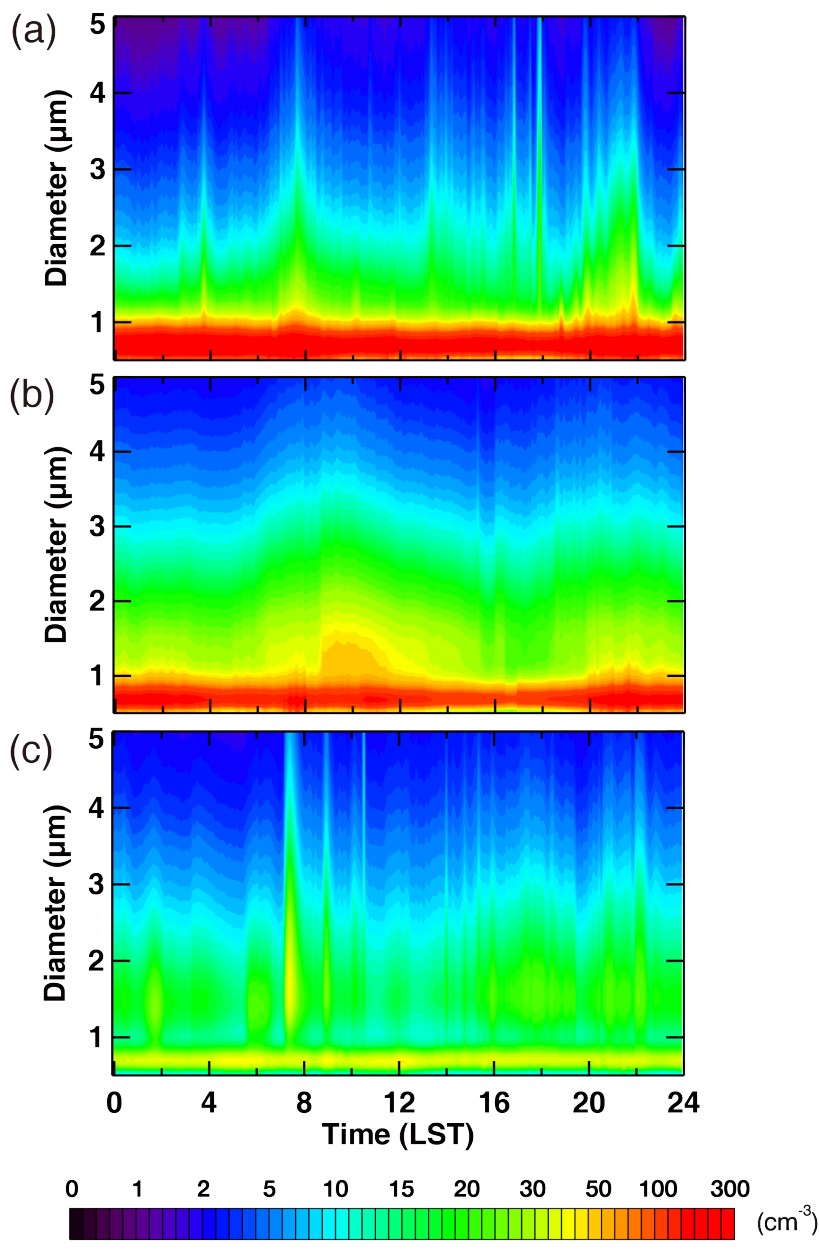

**Figure 6.** Same as **Figure 5** but for aerosol size distribution ($dN/dlogD_p$, 0.5 $\mu$m< $D_p$<5 $\mu$m) in **(a)** Wuwei, **(b)** Zhangye, and **(c)** Dunhuang from 3 April to 16 May 2014.



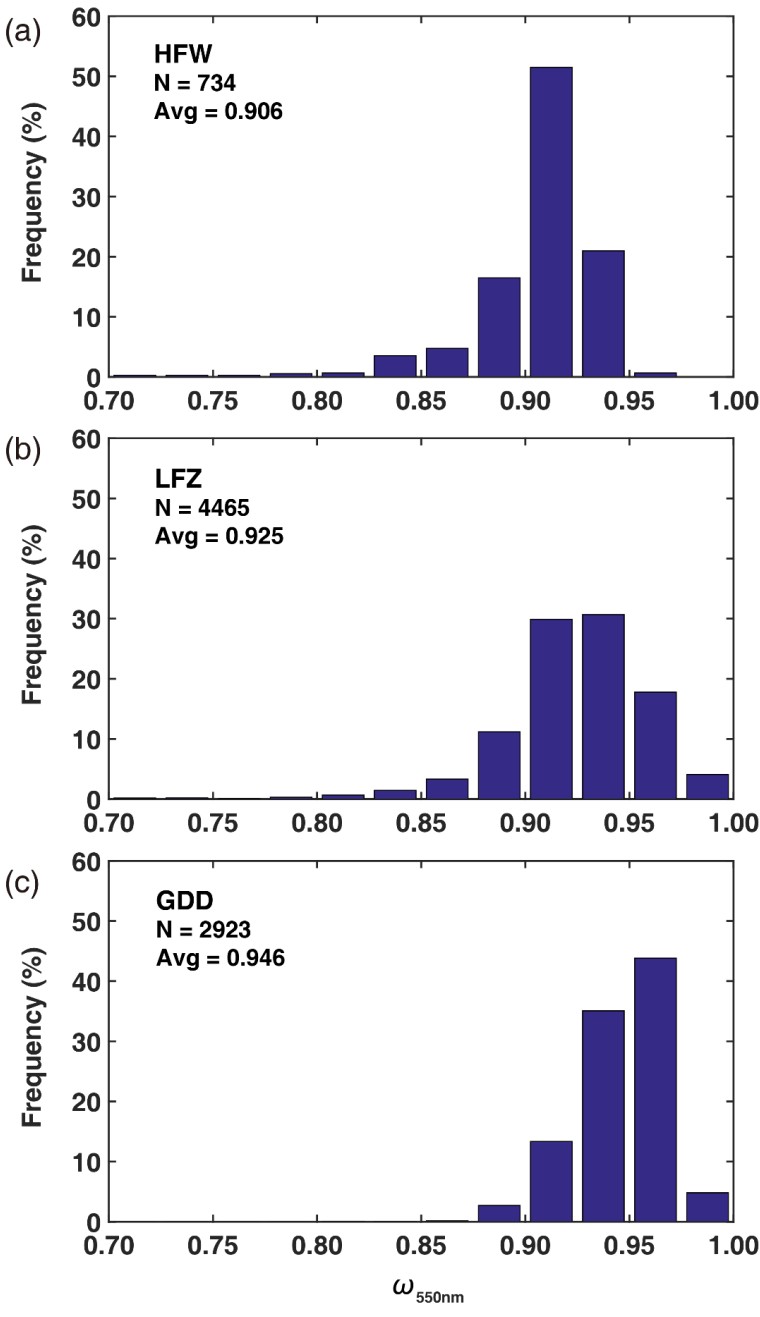

**Figure 7.** Histograms of single scattering albedo at 550 nm in **(a)** Wuwei, **(b)** Zhangye, and **(c)** Dunhuang. The numbers of samples and average values are also shown.





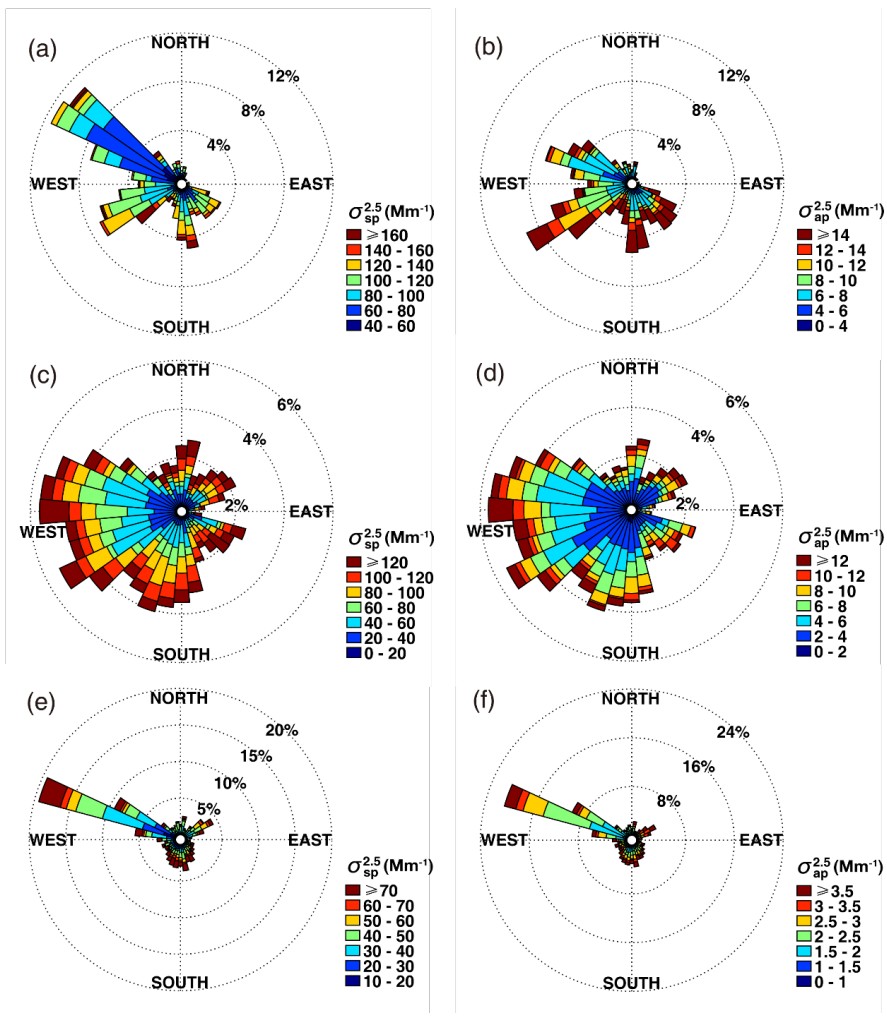

**Figure 8.** Wind roses for **(a)** aerosol scattering coefficient and **(b)** aerosol absorption coefficient at 550 nm in Wuwei; **(c)** and **(d)** are the same as **(a)** and **(b)** but for Zhangye; and **(e)** and **(f)** are the same as **(a)** and **(b)** but for Dunhuang. Note that data collected during the strong dust storm in Zhangye are excluded.





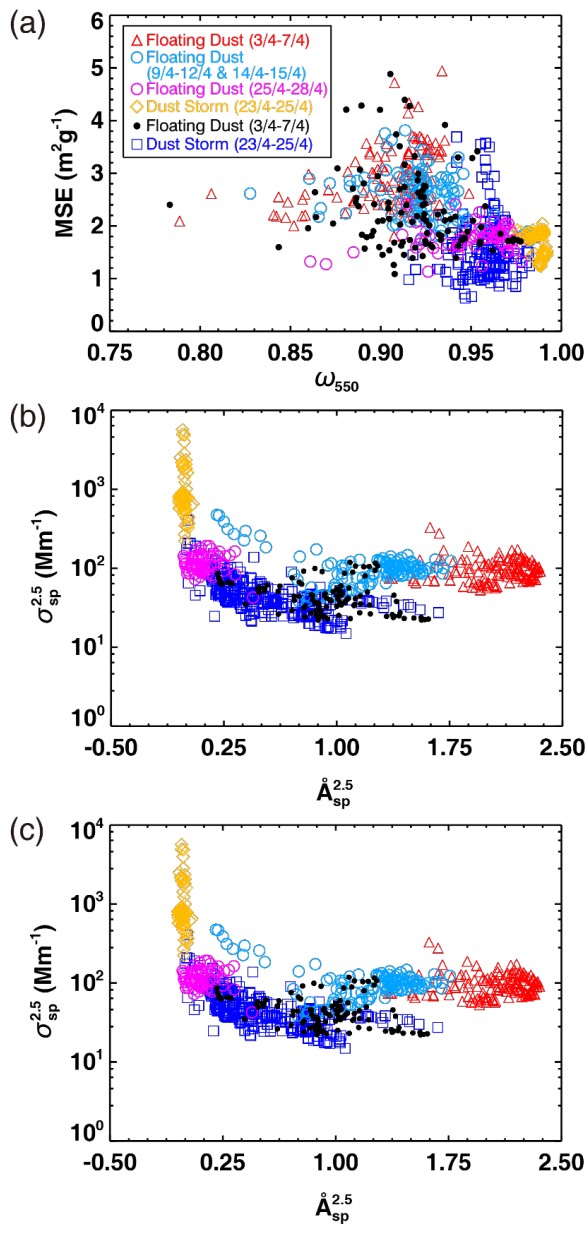

**Figure 9.** Scatter plots of **(a)** mass scattering efficiency (MSE) versus single scattering albedo ($\omega$) at 550 nm and **(b)** scattering Ångström exponent versus aerosol scattering coefficient for $PM_{2.5}$ at 550 nm; **(c)** is the same as **(b)** but for $PM_{1.0}$.



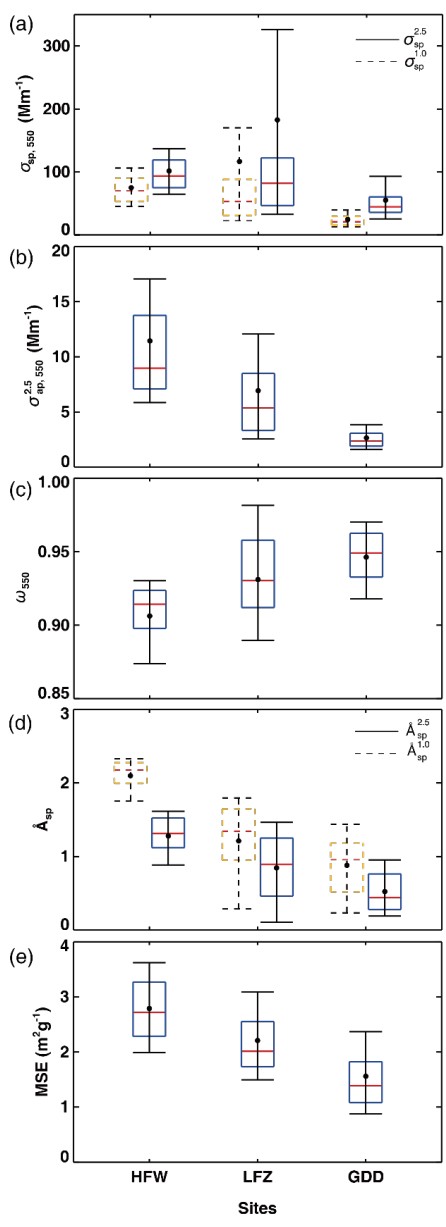

**Figure 10.** Box plots of (a) scattering coefficient, (b) absorption coefficient, (c) single scattering albedo ($\omega$) at 550 nm, (d) scattering Ångström exponent at 450-700 nm, and (e) mass scattering efficiency (MSE) at 550 nm during the dust field campaign. N represents the number of all in-run datasets in Wuwei (HFW), Zhangye (LFZ), and Dunhuang (GDD). The lines, moving from lower to upper, represent the 10th, 25th, 75th, and 90th percentiles. The red lines and black dots represent the median and mean values, respectively.





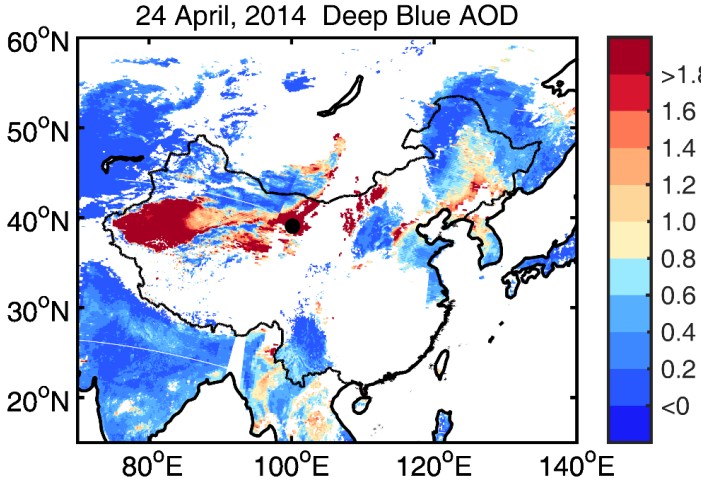

**Figure 11.** Terra MODIS Deep Blue AOD measured at 550 nm by the NASA Giovanni system during a heavy dust storm on 24 April 2014. The black dot represents the location of the ground-based mobile laboratory at Zhangye.





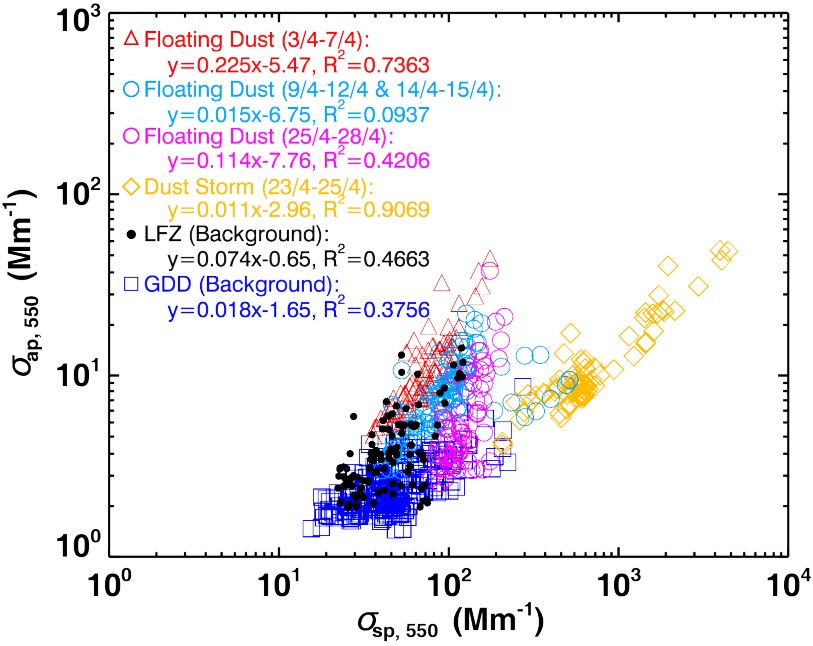

**Figure 12.** Scatter plot of aerosol absorption coefficients versus scattering coefficients from 3 April to 16 May 2014. The coloured symbols represent different atmospheric conditions during the dust field campaign.



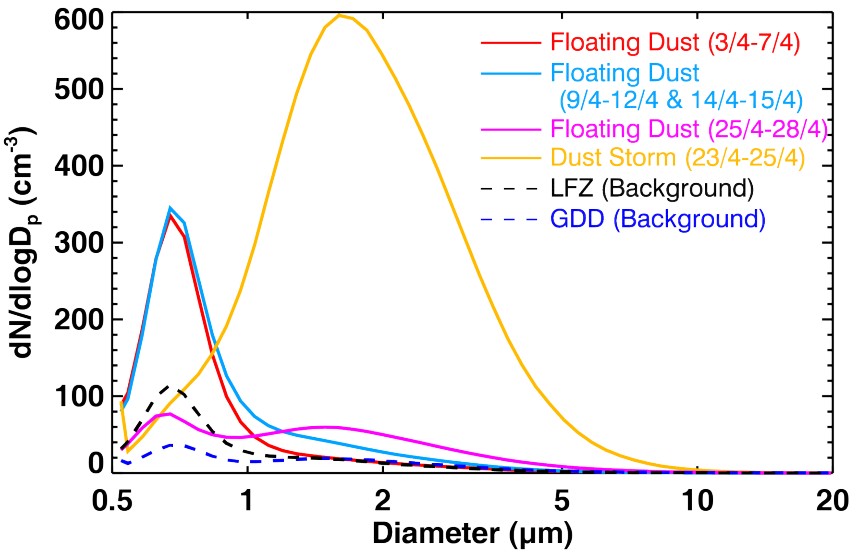

**Figure 13.** Average aerosol size distribution (dN/dlogD$_p$, cm$^{-3}$) based on all run data collected during the entire campaign.



**Table 1.** Statistics of optical properties of aerosols measured at three sites.

| | $\lambda$ (nm) | HFW | LFZ All | No dust | Dust storm | GDD |
|---|---|---|---|---|---|---|
| $\sigma_{sp}^{1.0}$ (Mm$^{-1}$) | 450 | 111.2±41.4 | 124.6±240.8 | 70.4±40.4 | 643.0±548.3 | 28.7±12.4 |
| | 550 | 74.9±29.1 | 116.6±260.3 | 57.1±33.6 | 686.2±589.7 | 23.8±11.9 |
| | 700 | 44.0±17.9 | 104.3±265.4 | 43.0±27.7 | 691.4±597.5 | 20.6±12.2 |
| $\sigma_{sp}^{2.5}$ (Mm$^{-1}$) | 450 | 132.8±49.8 | 193.4±432.1 | 98.2±63.2 | 1088.9±1009.9 | 58.4±33.0 |
| | 550 | 101.6±39.8 | 182.6±442.7 | 84.3±59.1 | 1108.0±1029.8 | 53.5±33.4 |
| | 700 | 75.7±32.0 | 169.2±440.4 | 70.6±55.3 | 1097.5±1018.7 | 48.5±33.0 |
| $\sigma_{ap}^{2.5}$ (Mm$^{-1}$) | 550 | 11.45±7.76 | 6.93±5.91 | 6.32±5.01 | 12.48±9.53 | 2.65±1.20 |
| $\omega$ | 550 | 0.906±0.029 | 0.931±0.038 | 0.925±0.035 | 0.987±0.004 | 0.946±0.021 |
| Å$_{1.0}$ | 450–700 | 2.10±0.24 | 1.05±0.67 | 1.18±0.57 | -0.15±0.05 | 0.88±0.44 |
| Å$_{2.5}$ | 450–700 | 1.28±0.29 | 0.77±0.52 | 0.86±0.48 | -0.02±0.02 | 0.53±0.32 |
| MSE (m$^2$ g$^{-1}$) | 550 | 2.79±0.68 | 2.21±0.78 | 2.26±0.79 | 1.72±0.23 | 1.55±0.73 |



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
