# Peer review of "Optical and microphysical properties of natural mineral dust and anthropogenic soil dust near dust source regions over northwestern China"

_Atmospheric Chemistry and Physics, 2017_

## Referee Comment (RC1) · Anonymous Referee #1 · 2 Oct 2017

This study presents the optical and physical properties of anthropogenic soil dust and natural mineral dust near the dust source regions in East Asia. This information is the key to evaluate the impacts of dust on the regional climate. Results and discussions are comprehensive, some valuable information have been generated. I would recommend the paper to be accepted for publication after a few comments as listed below have been addressed. 1. I strongly suggest the authors to reorganize the introduction section. The relevant studies on aerosol optical properties over East Asia should be reviewed. Accordingly, significance of this study could be further summarized and fo-

cus. For example, page 3 and page 4 all discussed the research importance of dust aerosol rather than their optical properties. 2. I suggest the authors to more pay attention to the logics between the sentences and paragraphs. For example, the logic in line 15-22 in page 3 is confusing. Organic matters and sulfate are the dominant chemical compositions of aerosol, why the authors only mentioned BC here? 3. I also strongly suggest authors to break down the result section into several topics or sections for reading friendly. 4. I suggest the authors to reorganize the abstract and conclusions due to these two sections are too similar. 5. The QA/QC of all instruments should be addressed in section 2.2. 6. Generally, the MAE of BC could be determined by its size distribution and coating. Why authors choose 6.6 m2 g-1?

———————————————————

---

## Referee Comment (RC2) · Anonymous Referee #2 · 1 Nov 2017

Wang et al.: Optical and microphysical properties of natural mineral dust and anthropogenic soil dust near dust source regions over Northwestern China, Atmos. Chem. Phys. Discuss., doi.org/10.5194/acp-2017-686, 2017

REVIEW

GENERAL

The paper presents measurements, results and analyses of optical properties and size distributions of surface layer aerosols in Northwestern China. Mineral dust affects air quality and climate over very large areas and they can be observed very far from their sources. In observations far from the sources the aerosol is typically aged and mixed with other particles. Therefore it is very valuable that measurements are conducted also near the sources. This manuscript presents measurements very close to or essentially at the source and is valuable as such. One of the weaknesses of the work is that particle size range of the optical measurements was limited to 2.5 µm. In dust storms there are often larger particles like the authors' own APS measurements show. But now the data are here and also they yield good information. The authors could use the full extent of the data to obtain also more information as I will suggest below.

I can recommed publishing the paper in ACP, but I did find something to be revised.

DETAILED COMMENTS

There is no text on the calibration of any of the instruments. In dusty conditions such as the sites where the measurements were conducted, instruments get quickly dirty and calibrations change. How did you deal with this? Write about calibrations, flow checks etc.

In addition to calibrations, also data processing needs some revision.
The nephelometer suffers from a problem called truncation which leads to underestimation of scattering. The error is the larger the larger the particles are. Read and cite Anderson and Ogren: Aerosol Sci. Tech., 29, 57–69, 1998 and Müller et al.: Aerosol Sci. Tech., 43, 581–586, 2009 and use their algorithms to correct the scattering. The corrected scattering coefficients will be larger than the ones presented now. And so will the corrected single scattering albedos also be.

The nephelometer used in the campaign also measures backscatter coefficient. Why is there nothing about that in the whole manuscript? It would be a valuable addition to the paper. If the instrument was working I strongly recommend presenting and discussing also backscatter coefficients and backscatter fractions at 3 wavelengths, both in figs and tables.

There was the APS. Why was that not utilized more? I have some suggestions, not requirements. First, calculating integrated volume concentrations for PM2.5 would yield some quality control when compared with the TEOM. And if they correlate well, they would together yield an estimate of the dust particle density, at least in such cases when particles were dominated by supermicron particles. That would be valuable.

Second: estimation of scattering coefficient would not be difficult either. If you don't have a Mie code, you can find them in the internet, calculate scattering efficiencies for the size channels of the APS and then calculate scattering coefficient of each size and finally integrate over the size range. An important question would be, for instance, how large a fraction of scattering did you not get measured because of the impactors in front of the nephelometers? Sure, the particles were not spherical and Mie theory not accurate but it would yield an estimate.

Third: the APS data could also be used for calculating some weighted mean diameter, e.g., volume-weighted mean diameter VMD of the size distribution and compare that with the Åsp. That would be valuable since satellite-derived products use wavelength dependency for estimating size.

There was also an SP2 in the campaign, at least according to Fig 3. Why was it and its data not discussed at all? It would potentially yield also interesting and important results. Comparison with MAAP in different cases for instance. The MAAP measures light absorption which may also be due to absorbing mineral aerosols, not just BC.

P6, L5 – What is Hexi corridor? Not well-known for non-Chinese.

P7, L3- Define or explain floating dust.

P8, L3-4, The detection limits of the scattering coefficients were obviously taken from the Table 4 of Anderson et al., 1996 for 300 min averaging time. But in that table there is not the multiplication by 10. So, the detection limit oftotal scattering at 450 nm is 0.44 Mm-1, not 0.44 x 10 Mm-1 like the authors claim on L3.

P8,L7: MAAP wavelength: the MAAP manual claims it is 670 nm but Müller et al. Atmos. Meas. Tech., 4, 245–268, 2011 measured it to be 637 nm. You should reprocess the data. First correct scattering for truncation, then use MAAP data for calculating SSA. But, instead of assuming the wavelength dependence of absorption, use the wavelength dependency (Åsp) of truncation-corrected scattering and interpolate the scattering to 637 nm and present SSA at 637 nm. This way you avoid assumptions. The point is that the wavelength dependency and Ångström exponent of absorption by absorbing mineral dust may significantly differ from 1.

In Fig 1, show
  - Hexi corridor – not well-known for most readers of ACP
  - show a kilometer scale also in the upper panel
  - use and show sub-panel letters a – d. Also for the upper panel.

---

## Author Response (AR1)

**This document includes a point-by-point response to the reviews, a list of all major changes made in the manuscript, and a marked-up manuscript version.**

**Response to Referee #1**

We are very grateful for the Referee #1's critical comments and suggestions, which have helped us improve the paper quality substantially. We have addressed all of the comments carefully as detailed below in our point-by-point responses. Our responses start with "R:".

General comments:

This study presents the optical and physical properties of anthropogenic soil dust and natural mineral dust near the dust source regions in East Asia. This information is the key to evaluate the impacts of dust on the regional climate. Results and discussions are comprehensive, some valuable information have been generated. I would recommend the paper to be accepted for publication after a few comments as listed below have been addressed.

R: We have addressed all of the comments carefully as detailed below.

I strongly suggest the authors to reorganize the introduction section. The relevant studies on aerosol optical properties over East Asia should be reviewed. Accordingly, significance of this study could be further summarized and focus. For example, page 3 and page 4 all discussed the research importance of dust aerosol rather than their optical properties.

R: We have reconstructed the introduction section and added one section in reviewing the aerosol optical properties over East Asia based on previous dust field campaigns.

I suggest the authors to more pay attention to the logics between the sentences and paragraphs. For example, the logic in line 15-22 in page 3 is confusing. Organic matters and sulfate are the dominant chemical compositions of aerosol, why the authors only mentioned BC here?

R: We agree with the reviewer. For this research mainly discuss the properties of natural and anthropogenic dust near the dust sources regions, we deleted the description of the other air pollutions in the introduction section, such as BC, OC, and sulfate aerosols.

I also strongly suggest authors to break down the result section into several topics or sections for reading friendly.

R: We have separated the result section into individual topics based on the reviewer's suggestion.

I suggest the authors to reorganize the abstract and conclusions due to these two sections are too similar.

R: The abstract has been rewritten, and the conclusions has been reorganized based on reviewer's suggestions.

The QA/QC of all instruments should be addressed in section 2.2.

R: The QA/QC information for all instruments have been given as Table 1, and the corresponding description is also added in Section 2.2.

Generally, the MAC of BC could be determined by its size distribution and coating. Why authors choose 6.6 m$^2$ g$^{-1}$?

R: Thanks very much for your comments and suggestions. To convert data into BC mass loadings, a precise knowledge of the mass absorption coefficient (MAC) is of great importance. Actually, a narrow range of BC for MAC (6.4–6.6 m$^2$ g$^{-1}$) was found to provide a good fit to urban particles collected by previous studies (Petzold et al., 1997; Penner et al., 1998; Sharma et al., 2002; Arnott et al., 2003; Bond and Bergstrom, 2006; Schwarz et al., 2008).

In this study, there are three reasons for us to use this MAC value as follows.

(1) Petzold et al. (2002) obtained the MAC= 6.5 ± 0.5 m$^2$ g$^{-1}$ at the wavelength of 670 nm for the black carbon particles using ambient aerosol samples.

(2) The MAAP (Model 5012) provides the absorption information as an equivalent black carbon concentration (EBC), which is obtained by dividing the measured absorption coefficient by a default MAC of 6.6 m$^2$·g$^{-1}$, recommended by the manufacturer.

(3) Müller et al. (2011) found that the optical wavelength of MAAP is 637 ± 1 nm instead of 670nm during the GAW2005 workshop. For an Ångström exponent of 1.02, the absorption coefficient at 637 nm should be 5% higher than that at 670 nm. Hence, the MAC at 637 nm is 5% lower and should be corrected by multiplication with a factor of 1.05, and the corrected equation was given by Müller: $\sigma_{ap,\ 637} = m_{BC} \cdot MAC \cdot 1.05$

But we also admit that the BC particles may tend to be mixed with other aerosols during aging process, and the MAC of BC can vary during its lifetime due to changes in its chemical composition. However, it must be noted that our research area is very close to the desert source regions. Therefore, as we used the same MAC value as the MAAP

recommended in our calculation, we consider that the variability in MAC as a source

of uncertainty can be neglected.

Above all, we prefer to use the MAC of BC as 6.6 $m^2$ $g^{-1}$ in this study.

**Table 1.** The main aerosol observations and ground-based instrumentations at three sites.

| Observation | Instrumentation | Model & manufacturer | Accuracy |
|---|---|---|---|
| Meteorological elements | Weather transmitter | WXT 520, Vaisala, Helsinki, Finland | $T$: $\pm$ 0.3; RH: 0.1 %; $P$: 0.1 hPa; WS: 0.1 m $s^{-1}$; WD: 1° |
| PM$_{2.5}$ concentration | Ambient particulate monitor | RP1400a, R&P Corp., Albany, NY, USA | 0.1 µg $m^{-3}$ |
| Aerosol total scattering/backscattering coefficient | Integrating nephelometer | TSI 3563, TSI Inc., Shoreview, MN, USA | 0.44, 0.17, and 0.26 $Mm^{-1}$ at the wavelengths of 450, 550, and 700 nm, respectively |
| Aerosol absorption coefficient | Multi-angle absorption photometer | MAAP 5012, Thermo Scientific, Waltham, MA, USA | 0.66 $Mm^{-1}$ |
| Aerosol size distribution | Aerodynamic particle sizer | APS 3321, TSI Inc., Shoreview, MN, USA | 0.001 $cm^{-3}$ |

**References**

Arnott, W. P., Moosmuller, H., Sheridan, P. J., Ogren, J. A., Raspet, R., Slaton, W. V., Hand, J. L., Kreidenweis, S. M., and Collett, J. L.: Photoacoustic and filter-based ambient aerosol light absorption measurements: Instrument comparisons and the role of relative humidity, J. Geophys. Res-Atmos, 108, 4034, doi:10.1029/2002jd002165, 2003.

Bond, T. C. and Bergstrom, R. W.: Light Absorption by Carbonaceous Particles: An Investigative Review, Aerosol Sci. Tech., 40, 27–67, doi:10.1080/02786820500421521, 2006.

Müller, T., Henzing, J. S., de Leeuw, G., Wiedensohler, A., Alastuey, A., Angelov, H., Bizjak, M., Collaud Coen, M., Engström, J. E., Gruening, C., Hillamo, R., Hoffer, A., Imre, K., Ivanow, P., Jennings, G., Sun, J. Y., Kalivitis, N., Karlsson, H., Komppula,

M., Laj, P., Li, S. M., Lunder, C., Marinoni, A., Martins dos Santos, S., Moerman, M., Nowak, A., Ogren, J. A., Petzold, A., Pichon, J. M., Rodriquez, S., Sharma, S., Sheridan, P. J., Teinilä, K., Tuch, T., Viana, M., Virkkula, A., Weingartner, E., Wilhelm, R., and Wang, Y. Q.: Characterization and intercomparison of aerosol absorption photometers: result of two intercomparison workshops, Atmos. Meas. Tech., 4, 245–268, doi:10.5194/amt-4-245-2011, 2011.

Penner, J. E., Chuang, C. C., and Grant, K.: Climate forcing by carbonaceous and sulfate aerosols, Clim. Dynam., 14, 839–851, doi:10.1007/s003820050259, 1998.

Petzold, A., Kopp, C. and Niessner, R.: The Dependence of the Specific Attenuation Cross-section on Black Carbon Mass Fraction and Particle Size, Atmos. Environ., 31, 661–672, doi:10.1016/s1352-2310(96)00245-2, 1997.

Petzold, A., Kramer, H., and Schonlinner, M.: Continuous measurement of atmospheric black carbon using a multi-angle absorption photometer, Environ. Sci. Pollut. R., 78–82, 2002.

Schwarz, J.P., Gao, R.S., Spackman, J.R., Watts, L.A. and Thomson, D.S.: Measurement of the Mixing State, Mass, and Optical Size of Individual Black Carbon Particles in Urban and Biomass Burning Emissions, Geophys. Res. Lett., 35, 13810–13814, doi:10.1029/2008gl033968, 2008.

Sharma, S., Brook, J.R. and Cachier, H.: Light Absorption and Thermal Measurements of Black Carbon in Different Regions of Canada, J. Geophys. Res.-Atmos, 107, 4771, doi:10.1029/2002jd002496, 2002.

**Response to Referee #2**

We greatly appreciate the Referee #2's insightful and constructive comments and suggestions, which are helpful and valuable for greatly improving our manuscript. We have addressed all of the comments carefully as detailed below in our point-by-point responses. Our responses start with "R:".

**General comments:**

The paper presents measurements, results and analyses of optical properties and size distributions of surface layer aerosols in Northwestern China. Mineral dust affects air quality and climate over very large areas and they can be observed very far from their sources. In observations far from the sources the aerosol is typically aged and mixed with other particles. Therefore, it is very valuable that measurements are conducted also near the sources. This manuscript presents measurements very close to or essentially at the source and is valuable as such. One of the weaknesses of the work is that particle size range of the optical measurements was limited to 2.5 μm. In dust storms there are often larger particles like the authors' own APS measurements show. But now the data are here and also they yield good information. The authors could use the full extent of the data to obtain also more information as I will suggest below.

I can recommend publishing the paper in ACP, but I did find something to be revised.

R: Thanks very much for your good suggestions and the acceptance of this work, we have addressed all of the comments carefully as detailed below.

**Detailed comments:**

There is no text on the calibration of any of the instruments. In dusty conditions such as the sites where the measurements were conducted, instruments get quickly dirty and calibrations change. How did you deal with this? Write about calibrations, flow checks etc.

R: We have added the details of the calibration and flow checks for all the aerosol-related instruments in Section 2.2, and the accuracy for each instrument is also listed in Table 1.

In addition to calibrations, also data processing needs some revision.

The nephelometer suffers from a problem called truncation which leads to underestimation of scattering. The error is the larger the particles are. Read and cite Anderson and Ogren: Aerosol Sci. Tech., 29, 57–69, 1998 and Müller et al.: Aerosol Sci. Tech., 43, 581–586, 2009. and use their algorithms to correct the scattering. The corrected scattering coefficients will be larger than the ones presented now. And so will the corrected single scattering albedos also be.

R: We are sorry for the misleading. In this study, the datasets of the aerosol optical properties have already been corrected based on the nonideal detection developed by Anderson and Ogren (1998), and one of the sentence has been added in Section 2.2 as "For reducing and quantifying the uncertainties in aerosol optical properties measured by the nephelometers, the data reduction and uncertainty analysis for the scattering datasets due to nonideal detection are followed by Anderson and Ogren (1998)."

The nephelometer used in the campaign also measures backscatter coefficient. Why is there nothing about that in the whole manuscript? It would be a valuable addition to the paper. If the instrument was working I strongly recommend presenting and discussing also

backscatter coefficients and backscatter fractions at 3 wavelengths, both in figures and tables.

R: Due to the backscatter coefficients shows the same trends with the total scattering coefficients but in a relatively small magnitude, we plotted a new figure suggested by the reviewer (Figure 4a). Then, we calculate the backscattering fractions at the wavelengths of 550 nm shown as Figure 4c in the revised manuscript. Additionally, the detailed information of backscatter coefficients and backscattering fractions of $PM_{1.0}$ and $PM_{2.5}$ at the wavelengths of 450, 550, 700 nm are listed in Table 2.

There was the APS. Why was that not utilized more? I have some suggestions, not requirements. First, calculating integrated volume concentrations for PM2.5 would yield some quality control when compared with the TEOM. And if they correlate well, they would together yield an estimate of the dust particle density, at least in such cases when particles were dominated by supermicron particles. That would be valuable.

R: Following the reviewer's suggestion, we found that the integrated volume concentrations of $PM_{2.5}$ measured by APS and the mass concentration measured by TEOM are correlated well as Figure S2 shown. Then, we calculated the dust particle density under different atmospheric conditions during the dust field campaign, and the relative discussion were added in the result section in Page 23, Line 2–9, and Table 3.

Second: estimation of scattering coefficient would not be difficult either. If you don't have a Mie code, you can find them in the internet, calculate scattering efficiencies for the size channels of the APS and then calculate scattering coefficient of each size and finally

integrate over the size range. An important question would be, for instance, how large a fraction of scattering did you not get measured because of the impactors in front of the nephelometers? Sure, the particles were not spherical and Mie theory not accurate but it would yield an estimate.

R: Thanks very much for your comments and suggestions. We use the Mie theory and the aerosol number size distribution measured by APS to estimate the scattering coefficient compared with that derived by the nephelometer. The real part of the refractive index was assumed to be 1.53, which was widely used for mineral dust in literatures (Müller et al., 2009; McConnell et al., 2010) the imaginary part of the refractive index was determined using Mie calculations. As shown in Figure 13, the Mie-calculated scattering coefficient and measured scattering coefficient are highly correlated. For instance, the imaginary part of the refractive index (0.0010) for natural dust during dust storm in Zhangye and the background weather condition in Dunhuang are similar to the result of SAMUM-1 in Saharan (Müller et al., 2009). Based on the Mie calculation in this study, the $PM_{2.5}$ scattering fraction, which defined as the contribution of the light scattering of $PM_{2.5}$ to the total scattering (the calculated scattering coefficient in the size range of 0.5–20 $\mu$m), is ~36.4 % during dust storm, while is in the range of ~37.9–85.1 % during floating dust episode. Detailed information of Mie-calculated and measured scattering coefficient is summarized in Table 3. Generally, most of the $\sigma_{sp, Mie}^{2.5}$ agree well with $\sigma_{sp, neph}^{2.5}$, which can reflect a good quality of the datasets of $\sigma_{sp}^{2.5}$ during this dust field campaign.

Third: the APS data could also be used for calculating some weighted mean diameter, e.g., volume-weighted mean diameter VMD of the size distribution and compare that with the

Åsp. That would be valuable since satellite-derived products use wavelength dependency for estimating size.

R: We use the APS data to calculate the volume-weighted mean diameter (VMD) under the diameter of 2.5 $\mu$m and 1.0 $\mu$m. We found that the $VMD_{2.5}$ and $Å_{sp}^{2.5}$ are correlated well during the whole dust field campaign (Figure S3 in Supplement). However, there is no significant linear correlation between $VMD_{1.0}$ and $Å_{sp}^{1.0}$. The highly possible explanation is that the $VMD_{1.0}$ is calculated based on the aerosol size diameter ranging from ~0.5 to 1 $\mu$m measured by APS, while the variation of $Å_{sp}^{1.0}$ is affected by the aerosol diameter under 1 $\mu$m.

There was also an SP2 in the campaign, at least according to Fig 3. Why was it and its data not discussed at all? It would potentially yield also interesting and important results. Comparison with MAAP in different cases for instance. The MAAP measures light absorption which may also be due to absorbing mineral aerosols, not just BC.

R: We feel sorry for the misleading. Yes, we also measure the BC concentration and its size distribution by using the SP2 instrument shown as Figure 3. But the major innovation of this manuscript is the difference of the optical and physical properties of natural and anthropogenic dust. Therefore, the datasets measured by SP2 are used to analyze the mixing status of BC with the other aerosols during this dust field campaign in another manuscript (In preparation). A comparison of the BC mass concentration between SP2 and MAAP instruments is given in Figure S1 in the Supplement. The result indicates that the tendency of BC mass concentrations are much similar, but the values measured by MAAP

was relatively larger than that measured by SP2. We note the relative large bias between MAAP and SP2 instruments may result from the size distribution of BC measured by using different sampler inlet impactors of 2.5 $\mu$m and 1 $\mu$m.

P6, L5 – What is Hexi corridor? Not well-known for non-Chinese.

R: The explanation of the geographical location of Hexi corridor is given in Section 2.1 based on the reviewer's suggestion as follow: "The Hexi Corridor is a ~1000km northwest-southeast-oriented chain of oases in northwestern China (mainly in the Gansu Province), surrounded by the Qilian Mountains (elevation: ~4000 m), the Beishan Mountains (elevation: ~2500 m), Heli Mountains (elevation: ~2000 m) and the Wushao Mountains (elevation: ~3000 m). The Hexi Corridor is considered to be a heavily polluted area because of the combination of local topography and the human activities occurring over northwestern China."

P7, L3- Define or explain floating dust.

R: The definition of floating dust has been added in Section 3.1 as "Floating dust is generally defined as a weather phenomenon in which fine mode dust particles suspended in the lower troposphere under calm or low-wind condition, with horizontal visibility less than 10 km."

P8, L3-4, The detection limits of the scattering coefficients were obviously taken from the Table 4 of Anderson et al., 1996 for 300 min averaging time. But in that table there is not the multiplication by 10. So, the detection limit of total scattering at 450 nm is 0.44 Mm$^{-1}$,

not 0.44 × 10 Mm$^{-1}$ like the authors claim on L3.

R: We have corrected this sentence as "the detection limits are 0.44 Mm$^{-1}$, 0.17 Mm$^{-1}$, and 0.26 Mm$^{-1}$ (1 Mm$^{-1}$ = 10$^{-6}$ m$^{-1}$), respectively" in Page 8, Line 20–21.

P8, L7: MAAP wavelength: the MAAP manual claims it is 670 nm but Müller et al. Atmos. Meas. Tech., 4, 245–268, 2011 measured it to be 637 nm. You should reprocess the data. First correct scattering for truncation, then use MAAP data for calculating SSA. But, instead of assuming the wavelength dependence of absorption, use the wavelength dependency (Å$_{sp}$) of truncation-corrected scattering and interpolate the scattering to 637 nm and present SSA at 637 nm. This way you avoid assumptions. The point is that the wavelength dependency and Ångström exponent of absorption by absorbing mineral dust may significantly differ from 1.

R: We have adjusted the absorption estimated by MAAP to 637nm following the method of Müller et al. (2011). Then, we interpolate the scattering coefficients to 637 nm in order to calculate SSA at 637 nm. We also replotted all of the related Figures as well as Table 2 based on the corrected datasets in the revised manuscript.

In Figure 1, show

- Hexi corridor – not well-known for most readers of ACP

R: The same with the above explanation.

- show a kilometer scale also in the upper panel

R: We have added the kilometer scale in the upper panel of Figure 1.

- use and show sub-panel letters a – d. Also for the upper panel.

R: We have added the sub-panel letters a–d in Figure 1.

[Figure]

**Figure S1.** Temporal variations of BC mass concentration under the diameter of 1 $\mu$m and 2.5 $\mu$m measured by SP2 and MAAP in Zhangye from 9 to 28 April, respectively.

[Figure]

**Figure S2.** Scatter plot of mass concentration of PM$_{2.5}$ versus the integrated volume concentration under the diameter of 2.5 $\mu$m during the dust field campaign. The color symbols represent different atmospheric conditions during the dust field campaign.

[Figure]

**Figure S3.** Scatter plot of the volume-weighted mean diameter (VMD) versus scattering Ångström exponent at 450–700 nm for **(a)** $PM_{2.5}$ and **(b)** $PM_{1.0}$. The color symbols represent different atmospheric conditions during the dust field campaign.

[revised manuscript text omitted]

under background conditions are ~21–163 Mm⁻¹, ~1.3–34.8 Mm⁻¹, and ~0.70–0.98, respectively. We note thatDuring a severestrong dust storm in Zhangye (i.e., from 23 to 25 April), the highestr values of $\sigma_{sp}^{2.5}$ (~5074 Mm⁻¹), backscattering coefficients ($\sigma_{bsp}^{2.5}$, ~522 Mm⁻¹) and $\omega$ (~0.993), the lowestr values of backscattering fraction ($b_{2.5}$, ~0.101) and $-\text{Å}_{sp}^{2.5}$ (~−0.046) at 450–700 nm, with peak values of aerosol number size distribution (appearing at the particle diameters range of 1–3 $\mu$m) indicate exhibit that the atmosphere aerosols particles observed during a strong dust storm in Zhangye were dominated by coarse mode dust aerosols. It is hypothesized that However,tThe relative higher values of MSE the highest during floating dust episodes in Wuwei and Zhangye are attributed to thereveal that the anthropogenic soil dust produced by agricultural cultivations. (e.g., land planning, ploughing, and disking)can scatter more solar radiation than coarse mode particles.

**1    Introduction**

The role of mineral dust aerosols (MDs) in the climate system has received considerable attention over recent years  (Arimoto et al., 2006; Ramanathan et al., 2001). MDs has a profound impact on the radiative balance of the Earth by scattering and absorbing solar radiation (Huang et al., 2010, 2014; Wang et al., 2010; Li et al., 2016); it can also act as cloud condensation nuclei (CCN) to alter the precipitation rate and hydrological cycle of the Earth (Rosenfeld et al., 2001). East Asia includes the Taklimakan, Tengger, Badain Jaran and Gobi Deserts and is thus considered to be one of the major source regions of natural dust in the world, as it produces large amounts of natural mineral dust (Zhang et al., 1997; Wang et al., 2008; Che et al., 2011, 2013; Ge et al., 2014; Xin, 2005, 2010, 2015).  The long-range transport of MDs from  dust source regions  have a significant influence on  aerosol radiative forcing and environment changing (Chen et al., 2013; Ge et al., 2011; Liu et al., 2015; Huang et al., 2008). In order to fully account the climate effects of MDs over eastern Asian regions, several international intensive field campaigns were conducted to measure their optical, physical, and chemical properties in recent decades, such as the Asian Aerosol Characterization

Experiment (ACE-Asia) (Arimoto et al., 2006), the NASA Global Tropospheric Experiment Transport and Chemical Evolution over the Pacific (TRACE-P) (Jacob et al.Christopher, 2003), the Atmospheric Brown Clouds-East Asia Regional Experiment (EAREX) (Nakajima et al., 20073), and the 2008 China-U.S. joint dust field experiment (Ge et al., 2010; Li et al., 2010; Wang et al., 2010). For instance, Ge et al. (2010) illustrated the mean single scattering albedo (SSA) measured at Zhangye over northwestern China increases with wavelength from $0.76 \pm 0.02$ at 415 nm to $0.86 \pm 0.01$ at 870 nm. Seasonal variations of the scattering coefficients and the absorption coefficients were also collected at Dunhuang and Zhangye of Gansu Province as well as Yulin of Shanxi Province,in northwestern China (Li et al., 2010; Xu et al., 2004; Yan et al., 2007). However, the systematic review of the optical and microphysical properties of the fine mode mineral dustMDs near the dust source regions in eastern Asia is still a challenge due to limited observations, especially for fine mode mineral dust near the dust source regionareas in northwestern China.

to measure their optical, chemical, and microphysical properties in East Asian regions over recent decades. For exampleinstance, the single scattering albedo, which is the measured of the effectiveness of scattering relative to total light extinction by aerosols, is a key variableparameter in assessing the climatic effects (Haywood and Shine, 1995). Dust aerosols are suggested to have SSA values typically greater than 0.95 in the visible wavelengths based on satellite reflectance and Aerosol Robotic Network (AERONET) data (Kaufman et al., 2001; Dubovik et al., 2002). A change in SSA from 0.9 to 0.8 can often change the sign of radiative forcing from negative to positive, which depends on

the reflectance of the underlying surface and the altitude of the aerosols (Hensen et al., 1997; Bergstrom et al., 2007). Therefore, the largest source of uncertainty in determining global radiative forcing is associated with the estimation of radiative forcing caused by dust particles, which is mainly due to their non-sphericity and chemical composition (Washington et al., 2003; McConnell et al., 2008,; IPCC, 2013). The Asian Aerosol Characterization Experiment (ACE-Asia) also showed that most of the coarse-particle nitrate and sulfate in post-frontal air was associated with dust (Arimoto et al., 2006). Dubovik and King (2000) pointed out that the average AERONET-retrieved single scattering albedo (SSA) is about 0.9 in the visible wavelength range at all stations, confirming the presence of aerosols with strong absorbing properties based on An International Regional Experiment (EAST-AIRE). The Asian Aerosol Characterization Experiment (ACE-Asia) showed that most of the coarse-particle nitrate and sulfate in post-frontal air was associated with dust, and more remarkably, that competitive or exclusionary processes evidently are involved in the uptake or production of these substances using an aerosol time-of-flight mass spectrometer (Arimoto et al., 2006). Nie et al. (2014) found observational evidence on new particle formation and growth in heavy dust plumes mixed with anthropogenic pollution and suggests an unexpected source of nucleating and condensable vapors via dust-induced heterogeneous photochemical processes based on field measurements at a mountain site in South China. Furthermore, the light-absorbing aerosols such as black carbon, organic carbon and desert dust in Asian monsoon regions may also induce dynamical feedback processes, leading to a strengthening of the early monsoon and

affecting the subsequent evolution of the monsoon (Li et al., 2016; Jin et al., 2015, 2016). The spatiotemporal distributions of dust physical and chemical properties (e.g., aerosol loading, SSA, asymmetry factor, and size distribution) were also measured at numerous sites havingwith different environmental conditions, especially during the EAST-AIRC (Li et al., 2011), Saharan Dust Expriment SHADE (Tanre et al., 2003), Puerto Rico Dust Experiment PRIDE (Reid et al., 2003), and 2008 China-U.S. joint dust field experiment (Ge et al., 2010; Li et al., 2010; Wang et al., 2010).

For instance, Washington et al. (2003) demonstrated that there isWe noted that the large-scale uncertainty is associated with the estimation of radiative forcing caused by dust particles, which is mainly due to their non-sphericity and chemical composition (Washington et al., 2003; McConnell et al., 2008). The single scattering albedo, which is the measured of the effectiveness of scattering relative to total light extinction by aerosols, is a key variable in assessing the climatic effects (Haywood and Shine, 1995). Dust aerosols are suggested to have SSA values typically greater than 0.95 in the midvisible wavelengths based on satellite reflectance and Aerosol Robotic Network (AERONET) data (Kaufman et al., 2001; Dubovik et al., 2002). Lower SSA values (~0.8 –0.9) were expected in the source regions for Asian dust particles since they are often mixed with anthropogenic light-absorbing aerosols during transport over the downwelling industrial and urban areas (Kim, 2001; Sohn et al., 2007). Ge et al. (2010) also found similar values of SSA from $0.76 \pm$ +-0.02 at the wavelength of 0.415 $\mu$um to $0.86 \pm$ $\mu$-0.01 at the wavelength of 0.870 $\mu$um at Zhangye over northwestern China. A change in single scattering albedo (SSA) from 0.9 to 0.8 can often change the sign of

radiative forcing from negative to positive, which depends on the reflectance of the underlying surface and the altitude of the aerosols (Hensen et a., 1997; Bergstrom et al., 2007). Compared to natural mineral dust, black carbon (BC), which is generated from the incomplete combustion of fossil fuels and biomass burning, is also a major

5 anthropogenic pollutant. Numerous modelling studies have demonstrated the importance of BC in regional climate change (Ramanathan et al., 2005; Menon et al., 2002). Therefore, the largest source of uncertainty in determining global radiative forcing is the quantification of the direct and indirect effectsoptical and microphysical properties of mineral dustMDs and other anthropogenic aerosols (IPCC,

10 2013).

Recently, the potential impacts of anthropogenic soil dust have also received an increasing amount of attention (Prospero et al., 2002; Huang et al., 2015a; Tegen and Fung, 1995; Tegen et al., 2002; Shi et al., 2015; Pu et al., 2015; Wang et al., 2015a). Anthropogenic mineral dust can also influence air quality and human health through

15 the processes of their emission, transport, removal, and deposition (Aleksandropoulou et al., 2011; Chen et al., 2013; Huang et al., 2014, 2015a, 2015b; Kim et al., 2009; Li et al., 2009; Mahowald and Luo, 2003; Zhang et al., 2005, 2015). Ginoux et al. (2010) estimated that anthropogenic dust accounts for 25% of all dust aerosols using observational data from MODIS (Moderate Resolution Imaging Spectroradiometer)

20 Deep Blue satellite products combined with a land-use fraction dataset. Nie et al. (2014) found observational evidence on new particle formation and growth in heavy dust plumes mixed with anthropogenic pollution via dust-induced heterogeneous

photochemical processes. northeastern China and its surrounding regions are generally

5  Because anthropogenic dust emissions from disturbed soils are not well constrained, we define anthropogenic dust as mineral dust from areas that have been disrupted by human activities, such as deforestation, overgrazing, and agricultural and industrial activities (Aleksandropoulou et al., 2011; Tao et al., 2014, 2015, 2017; Tegen and Fung, 1995;

10 Tegen et al., 2002, 2004; Thompson et al., 1988); anthropogenic dust is thus different  from natural mineral dust, which originates from desert regions (Che et al., 2011, 2013; Goudie and Middleton, 2001; Li et al., 2012; Park and Park, 2014; Pu et al., 2015; Wang et al., 2008, 2010). This assumption is consistent with the results of a recent study by Huang et al. (2015a), who found that anthropogenic dust comprises 91.8% of

15 regional emissions in eastern China and 76.1% of regional emissions in India

Understanding the  natural dust mixed with

20 the  anthropogenic aerosols  in the troposphere has a critical impact on our ability to get insight into atmospheric compositions and predict global climate change (Nie et al., 2014; Ramanathan et al.,

2007; Spracklen and Rap, 2013; Wang et al., 2015b). Although sSeveral attempts have been made conducted to investigate the significance of the effects of dust on global climate, meteorology, atmospheric dynamics, ecosystems and human health (Rosenfeld et al., 2011; Qian et al., 2004), only lLimited field campaigns have focused on the properties of natural dust and anthropogenic aerosols, especially those of the anthropogenic dust aerosols produced by human activities near dust source regions. In this paper, In this study, we not only focus on the surface measurements of theed optical and microphysical properties of natural dust and anthropogenic aerosolsdust. This paper presents detailed emission information obtained from measurements made in Wuwei, Zhangye, and Dunhuang over Northwestern northwestern China from 3 April to 16 May in 2014, but also in order to better understand the sources of regional emissions and the mixing state of air pollution with mineral dust. We also used statistical analysis to identify the possible signatures of natural dust storms transported from dust source regions over northwestern China.

**2    Methodology**

**2.1    Measurement Sites dDespcription**

The Hexi Corridor is a ~1000km northwest-southeast-oriented chain of oases in northwestern China (mainly in the Gansu Province), surrounded by the Qilian Mountains (elevation: ~4000 m) to the south and southwest, the Beishan Mountains (elevation: ~2500  m), and Heli Mountains (elevation: ~2000 m) to the north, and the Wushao Mountains (elevation: ~3000 m) to the east.; adjacent to the east margin of the

Kumtag Desert, by the Badain Jaran Desert to the northeast and the Tengger Desert to the southeast. 
[revised manuscript text omitted]

**3 Results**

**3.1 Temporal Variability RegionalGeneral Statistics**

Floating dust is generally defined as a weather phenomenon in which fine mode dust particles suspended in the lower troposphere under calm or low-wind condition, with horizontal visibility less than 10 km; while dust storm is that large quantities of dust particles lofted by strong winds, and horizontal visibility reduced to below 1 km.

(Wang et al., 2005; Wang et al., 2008). During this dust field campaign,  three floating dust episodes (which are shown as dotted boxes in Figure 4) occurred on 3–7 April in Wuwei and on 9–12 and 25–28 April in Zhangye. We also observed the optical and microphysical properties of natural mineral dust during a heavy dust storm

5 (shown as a solid box in Figure 4) from 23 to 25 April  in Zhangye. Moreover, we identified five clear-sky days in Zhangye (16, 18, 19, 20 and 22 April) and three clear-sky days in Dunhuang ( 11, 14, and 15 May) as background weather conditions based on the manual weather recording and the abovementioned measurements. According to the land surface types shown in Figure 2, one of the major

10  novelties of this study is to investigate the characteristics of anthropogenic and natural dust during floating dust  and dust storms episodes, respectively

15

Figure 4 illustrates the temporal variations  of hourly averaged $\sigma_{sp}$, $\sigma_{bsp}$, , $\sigma_{ap}$ , $b$, $\omega$ , as well as $\text{Å}_{sp}$  MSE,  and aerosol size distribution in

20 Wuwei, Zhangye, and Dunhuang in chronological order from 3 April to 16 May 2014.   Note that the time periods denoted in Figure 4 contain

some gaps due to the transportation of ground-based mobile facility or instrument problems. The statistical analyses of the optical parameters  are also summarized in Table 2. Hereinafter, these results are given as the mean $\pm$ the standard deviation of the hourly averaged datasets. Unless otherwise noted, all aerosol scattering measurements discussed here are for the wavelength of 550 nm.

Aerosol optical and microphysical properties are entirely different in these three sites. One of the most significant features in Figure 4a is that the variation of $\sigma_{sp}^{2.5}$ is highly consistent with that of $\sigma_{sp}^{1.0}$ during the whole period of field campaign; the backscatter coefficients shows the same trends with the total scattering coefficients but in a relatively small magnitude. The values of $\sigma_{sp}^{2.5}$ and $\sigma_{sp}^{1.0}$ are very close in Wuwei and Zhangye; however, the large differences observed in Dunhuang. It indicates that fine mode particles dominate the scattering coefficient in farmland regions, whereas coarse mode particles play a more important role in  the desert regions .  Meanwhile, the large standard deviations are found in Wuwei and Zhangye, which are possibly attribute to frequent floating dust events and local anthropogenic emissions (Wang et al., 2008, 2015). Except for the values obtained during a heavy dust storm , the hourly mean $\sigma_{sp}^{2.5}$ ~~values measured at the wavelength of 550 nm using the nephelometer range from ~ 50  429 20  532atof~~in

Wuwei and Zhangye, respectively; the corresponding $\sigma_{bsp}^{2.5}$ are 12.2 ± 4.4 Mm⁻¹ and

9.5 ± 5.9 Mm⁻¹. By contrast, the much lower $\sigma_{sp}^{2.5}$ (54.0 ± 32.0 Mm⁻¹) and $\sigma_{bsp}^{2.5}$

(6.5 ± 3.7 Mm⁻¹) are measured in Dunhuang. Values for . The *b* increased with the

wavelengths due to decreasing size parameters. with the $b_{2.5}$ ofare 0.121 ± 0.005 and,

0.00115 ± 0.007 and 0.122 ± 0.005 in Wuwei, Zhangye, and Dunhuang, respectively,

which are consistent with the result fromin Backgarden (0.124 ± 0.015; Garland et al.,

2008), a rural site near the megacity Guangzhou in southeastern China, which can be

compared to the $\sigma_{ap}$ values at 637 nm of 9.7 ± 6.0 3.6~69 Mm⁻¹ and 5.5 ± 3.8~1.3~

64.5 Mm⁻¹ measured at Wuwei and Zhangye, respectively. , but higher than those

observed in Shouxian in eastern China (0.101 ± 0.017; Fan et al., 2010).

Meanwhile, the large standard deviations of $\sigma_{ap}^{2.5}$ are found in Wuwei and Zhangye,

which are possibly attribute to frequent floating dust events and local anthropogenic

emissions (Wang et al., 2008, 2015a). The lowest value of $\sigma_{ap}^{2.5}$ $\sigma_{ap}$ obtained during the

field campaign (i.e., 2.7 3 ± 1.20.9 Mm⁻¹) was are collect obtainedmeasured in

Dunhuang, which can be compared with the relative higher $\sigma_{ap}$ values of 9.7 ± 6.0

Mm⁻¹ and 5.5 ± 3.8 Mm⁻¹ in Wuwei and Zhangye, respectively. This observation

reveals that natural mineral dust is still a weaker absorber than anthropogenic soil dust

that has been mixed with air pollutants (e.g., BC and OC).

Compared with Figure 4a4b, Figure 4b 4d indicates that the majority of $\omega_{550}\omega_{637}$

values are much higher in Dunhuang than they arethat in the other two sites;, where

these values range from ~0.800.874 to 0.99986, with overalla mean value of

[revised manuscript text omitted]
 in Zhangye (i.e., 3–7 9–12, 14–15, 25–28 April), For instance, tThe average values of $\sigma_{sp}^{2.5}$ at 550 nm, $\sigma_{bsp}^{2.5}$, $\sigma_{ap}^{2.5}$, and $\omega_{550 637}$ at 550 nm at 637 nm, $\text{Å}_{sp}^{2.5}$, and MSE and d$N$/dlog$D_p$ are $102 \pm 37$ Mm$^{-1}$, $12.2 \pm 4.4$ Mm$^{-1}$, $9.7 \pm 6.1$ Mm$^{-1}$, $0.902 \pm 0.025$, $1.28 \pm 0.27$ and $2.79 \pm 0.57$ m$^2$ g$^{-1}$, respectively; during two floating dust periods in Zhangye (i.e., 9–12 and 25–28 April), the corresponding values are $183$ $118$ $5 \pm 64$ $36$ Mm$^{-1}$, $12.1 \pm 4.2$ Mm$^{-1}$, $6.9$ $84 \pm 3.7$ $4.0$ Mm$^{-1}$, $0.935 \pm 0.030$ $0.91$, $0.90.81$ $73 \pm 0.59$ $7$, and $2.4$ $3$ $24 \pm 0.57$ m$^2$ g$^{-1}$ and $2072 \pm 795$ cm$^{-3}$, respectively, during floating dust periods in Zhangye (i.e., 9–12, 14–15, 25–28 April); whereas these values are sequentially $1108$ $1088 \pm 991$ Mm$^{-1}$,

114.6 ± 101.6 Mm$^{-1}$, 13.510.6 ± 7.6 Mm$^{-1}$, 0.99989 ± 0.004, −0.02014 ± 0.018 , and

1.73 ± 0.20 m$^2$ g$^{-1}$ , and 9923±2698 cm$^{-3}$ , respectively, during dust storms in Zhangye

(i.e., 23–25 April). The number concentrations size distribution (d$N$/dlog$D_p$) of coarse

mode particles with diameters of 1–5 3 $\mu$m can reach a peak of ~900 590 cm$^{-3}$, which

reveals that pure coarse mode particles from desert regions were dominant during dust

storms in Zhangye. However, the overall variations of in $\omega_{637}\omega_{550}$, which ranges from

~0.7182 to 0.95 and ~0.6483 to 0.98 during floating dust episodes in Wuwei and

Zhangye, respectively, indicate that atmospheric aerosols not only include

anthropogenic soil dust that is smaller than 1 $\mu$m but has also undergone mixing with

air pollutants (e.g., BC) because of their transportation from urban and industrial

regions. In addition to the large discrepancies between fine mode and coarse mode

particles observed under different weather conditions, there are also significant

differences between the optical and microphysical properties of the given atmospheric

particles that are smaller than 1 $\mu$m and 2.5 $\mu$m. We note that the values of $\sigma_{sp}^{1.0}$

(75 74 ± 29 27 Mm$^{-1}$) are only slightly lower than those of $\sigma_{sp}^{2.5}$ (102 101 ± 40 37 Mm$^{-1}$) that are observed in Wuwei. However, there are significant differences between the

values of Å$_{sp}^{1.0}$ and Å$_{sp}^{2.5}$, which range from ~1.1–2.4 (mean: 2.1) and ~0.2–1.7 (mean:

1.3)are 2.1 ± 0.2 and 1.3 ± 0.3, respectively, because of the occurrence of floating dust

episodes in Wuwei.

By We usinged a modified Mie theory and the aerosol number size distribution

measured by APS to estimate the scattering coefficients. The imaginary parts of the

refractive indexes (0.0010) fuor natural dust during dust storm in Zhangye and the

background weather condition in Dunhuang are 0.001, while the higher value of imaginary part (0.01~0.08) during floating dust reflects inherently more anthropogenic dust particles, which reflects

5       that these atmospheric

10 aerosols during this dust field campaign not only were dominated by anthropogenic soil dust produced by agricultural cultivations in Wuwei and Zhangye, but also underwent  natural mineral dust originated from the dust source regions over northwestern China.

15

**5    Data availability**

All data sets and codes used to produce this study can be obtained by contacting Xin

20 Wang (wxin@lzu.edu.cn). The MODIS data used in this study are available at Aerosol Product, https://modis.gsfc.nasa.gov/data/dataprod/mod04.php.

*Competing interests.* The authors declare that they have no conflict of interest.

*Acknowledgements.* This research was supported by the Foundation for Innovative

5    Research Groups of the National Science Foundation of China (41521004), the

National Science Foundation of China under Grant (41775144 and 41522505), and the

Fundamental Research Funds for the Central Universities (lzujbky-2015-k01 and

lzujbky-2016-k06). The MODIS data were obtained from the NASA Earth Observing

System Data and Information System.

[Figure]

**Figure 1.** (a) The sampling locations of ground-based mobile laboratory and their

surrounding areas near dust source regions during the 2014 dust field campaigns at (**b**) Gobi Desert in Dunhuang (GDD, 39.96°N, 94.33°E; 1367 m a.s.l.), (**c**) Linze Farmland in Zhangye (LFZ, 39.04°N, 100.12°E; 1578 m a.s.l.) and (**d**)  Huangyang Farmland in Wuwei (HFW, 37.72°N, 102.89°E; 1691 m a.s.l.).

[Figure]

**Figure 2.** Same as **Figure 1** but for land surface conditions at **(a)** Huangyang Farmland in Wuwei (HFW), **(b)** Linze Farmland in Zhangye (LFZ), and **(c)** Gobi Desert in Dunhuang (GDD).

[Figure]

[Figure]

[Figure]

**Figure 3. (a)** The ground-based mobile laboratory in Dunhuang and **(b)** the schematic diagram of the ensemble instrumentation system.

[Figure]

[Figure]

**Figure 4.** Temporal variations  of hourly -averaged **(a)** aerosol scattering  coefficient at 550 nm and absorption coefficient at 637nm, **(b)** the backscattering fractions at 550 nm **(c)** single scattering albedo at  637 nm, **(d)** scattering Ångström exponent (calculated from 450 nm to 700 nm), **(e)** mass scattering efficiency (MSE) of PM$_{2.5}$ at 550 nm, and **(f)** aerosol size distribution (d$N$/dlog$D_p$ , 0.5 $\mu$m < $D_p$ < 5 $\mu$m) during the entire period from 3 April to 16 May 2014. The shaded box represents a strong dust storm that occurred in Zhangye, and the dotted boxes represent  three floating dust episodes that occurred in Wuwei and

Zhangye.

[Figure]

[Figure]

**Figure 5.** Diurnal variations in **(a)** aerosol scattering coefficient at 550 nm, where solid lines represent the variations  of PM$_{2.5}$ and dotted lines represent the variations  of PM$_{1.0}$; **(b)** the aerosol absorption coefficient at 637 nm and the scattering Ångström exponent for **(c)** PM$_{2.5}$ and **(d)** PM$_{1.0}$ (both calculated from 450  to 700 nm); **(e)** single scattering albedo at  637 nm; and **(f)** mass scattering efficiency (MSE) at 550 nm in Wuwei, Zhangye, and Dunhuang from 3 April to 16 May 2014. Note that data collected during the strong dust storm in Zhangye are excluded.

[Figure]

[Figure]

**Figure 6.** Same as **Figure 5** but for aerosol size distribution ($\mathrm{d}N/\mathrm{dlog}D_\mathrm{p}$ $_\mathrm{p}$, 0.5 $\mu$m $< D_\mathrm{p}$$_\mathrm{
[revised manuscript text omitted]